# Modified Nano-TiO$_2$ Based Composites for Environmental Photocatalytic Applications

**Shuo Wang [1], Zhu Ding [1], Xue Chang [1], Jun Xu [2]**  **and Dan-Hong Wang [1,3,*]**

[1] TKL of Metal and Molecule Based Material Chemistry, National Institute for Advanced Materials, School of Materials Science and Engineering, Nankai University, Tianjin 300350, China; wshuo@mail.nankai.edu.cn (S.W.); dingzhu91@mail.nankai.edu.cn (Z.D.); changxuenku@126.com (X.C.)

[2] Center for Rare Earth and Inorganic Functional Materials Tianjin Key Lab for Rare Earth Materials and Applications, School of Materials Science and Engineering and National Institute for Advanced Materials, Nankai University, Tianjin 300350, China; junxu@nankai.edu.cn

[3] Key Laboratory of Advanced Energy Materials Chemistry (Ministry of Education), College of Chemistry, Nankai University, Tianjin 300071, China

\* Correspondence: dhwang@nankai.edu.cn; Tel.: +86-138-2112-7707

**Abstract:** TiO$_2$ probably plays the most important role in photocatalysis due to its excellent chemical and physical properties. However, the band gap of TiO$_2$ corresponds to the Ultraviolet (UV) region, which is inactive under visible irradiation. At present, TiO$_2$ has become activated in the visible light region by metal and nonmetal doping and the fabrication of composites. Recently, nano-TiO$_2$ has attracted much attention due to its characteristics of larger specific surface area and more exposed surface active sites. nano-TiO$_2$ has been obtained in many morphologies such as ultrathin nanosheets, nanotubes, and hollow nanospheres. This work focuses on the application of nano-TiO$_2$ in efficient environmental photocatalysis such as hydrogen production, dye degradation, CO$_2$ degradation, and nitrogen fixation, and discusses the methods to improve the activity of nano-TiO$_2$ in the future.

**Keywords:** nano-TiO$_2$; photocatalytic applications; visible light; doping; vacancy; composite

## 1. Introduction

Fossil fuel is a non-renewable resource with limited reserves [1–3]. Over the past 140 years, we have consumed one trillion barrels of oil, and today, the world's demand for energy has exceeded 1000 barrels of oil, 100,000 cubic meters of natural gas, and 221 tons of coal per second [4]. It is clear that the consumption of fossil fuels is not sustainable in the long-term and has caused serious harm to the environment and human health [5]. It is reported in [6] that the energy challenges we face are related to the "tragedy of the commons": we treat fossil fuels as resources that can be extracted and used in any way by anyone, anywhere, and use the Earth's atmosphere and oceans as landfills, emitting more than 30 Gt of CO$_2$ per year [7].

Dyes play an important role in various sectors of the dyeing and textile industries, most of which are synthetic dyes. These dyes usually come from coal tar and petroleum intermediates, with the annual output of more than $7 \times 10^5$ tons [8–10]. Konstantinou and Albanis [11] mentioned that industrial and textile dyes are mostly toxic organic compounds. It is estimated that nearly 17 to 20 percent of water pollution is related to the textile finishing and dyeing industries [12]. In 1974, the Ecological and Toxicological Association of the Dyestuffs Manufacturing Industry (ETAD) was established to protect consumers by working fully with the government to minimize environmental damage by addressing issues related to the toxicological impact of the products [13]. In the ETAD survey, among the 4000 test samples, the LD$_{50}$ value of 90% of dyes was more than $2 \times 10^{-3}$ mg kg$^{-1}$. Of all the tested dyes, diazo,

direct, and alkaline dyes showed the highest toxicity [14]. In addition to the textile industry, hair dye [15], the leather industry [16], paper industry [17], photochemical batteries [18], and luminescent solar concentrator (LSC) technologies [19–21] also use a large amount of dyes.

In order to effectively control environmental pollution and solve energy problems, it is an important task for researchers to develop efficient, stable, and green solutions. Solar energy is inexhaustible and its annual radiation energy to the surface of the Earth equivalent to about 140 trillion tons of coal burning energy. It belongs to the clean renewable pollution-free energy group and is thus the most promising energy to promote the rapid development of mankind in the future [22].

In 1972, Japanese scientists Fujishima and Honda [23] were the first to demonstrate photocatalytic splitting of water on a $TiO_2$ electrode. Their work promotes the semiconductor photocatalysis widely used in the field of environment and energy, which opened the door of photocatalysis and has since attracted the interest of a large number of scientific researchers. In recent years, the research and development of a nano-$TiO_2$ (nano-$TiO_2$ generally refers to particles smaller than 100 nm in at least one dimension [24–26]) photocatalyst has developed rapidly and more and more achievements have been made.

For $TiO_2$, as the particle size decreases, its photocatalytic activity will increase to a certain extent, showing a specific size effect. Taken together, the possible size effects of nano-$TiO_2$ photocatalytic materials are as follows:

(1) Quantum size effect: $TiO_2$ is an n-type semiconductor [27]. When its particle size is less than 50 nm, it will have different properties from single crystal semiconductors, which is called the "quantum size effect" [28]. That is, when the particle size drops to a certain value, the electron energy level near the Fermi level changes from a quasi-continuous to a discrete energy level or a widening energy gap. At this time, the potential of the conduction band becomes more negative, and the potential of the valence band becomes more positive, thereby increasing the energy of photogenerated electrons and holes, enhancing the redox capability of the semiconductor photocatalyst and improving its photocatalytic activity [29–31].

(2) Surface area effect: First, as the particle size decreases to the nanometer, the specific surface area of the photocatalyst will greatly increase as well as the number of surface atoms, so that the light absorption efficiency will be improved and the surface photocarrier concentration will increase accordingly, so the surface redox reaction efficiency is thus improved [32]. Second, as the particle size decreases, the specific surface area increases, and the bonding state and electronic state of the surface are different from the inside. The unsaturated coordination of surface atoms leads to an increase in surface active sites. Therefore, compared with powders of large particle sizes, their number of surface active sites are higher, so the adsorption capacity of the substrate is enhanced, and the reaction activity is increased [33]. In addition, the number of hydroxyl groups on the surface of the catalyst directly affects the catalytic activity during the photocatalytic reaction. When $TiO_2$ powder is immersed in the aqueous solution, the surface undergoes a hydroxylation process. Therefore, as the size decreases, the specific surface area increases, and the number of surface hydroxyl groups also increases, thereby improving the reaction efficiency [34,35].

(3) Carrier diffusion effect: The grain size also has a great influence on the recombination rate of photogenerated carriers. For nanoscale semiconductor particles, the particle size is usually smaller than the thickness of the space charge layer, and any effect of the space charge layer can be ignored [36]. The smaller the particle, the shorter the time for photogenerated electrons to diffuse from the crystal to the surface, and the lower the probability of electron and hole recombination in the particle will be, which improves the photocatalytic efficiency [37–39].

There have been many outstanding studies on the synthesis and modification of $TiO_2$ based photocatalysts and their applications in solving energy and environmental problems [40–42]. However, few articles have classified the environmental application of nano-$TiO_2$ and compared the specific applications of nano-$TiO_2$ based composites in various classifications [43–46]. By doping metal and nonmetal ions, introducing vacancy and fabricating composites with other semiconductors, the band

gap of TiO$_2$ was adjusted to make it have better photocatalytic activity [47]. In this review, we intend to summarize the applications of nano-TiO$_2$ in environmental photocatalysis such as hydrogen production, carbon dioxide degradation, dye degradation, and nitrogen fixation. Finally, the current challenges and key issues of the nano-TiO$_2$ photocatalyst are described, which need to be addressed in future research.

## 2. Titanium Dioxide: An Introduction

### 2.1. TiO$_2$ Structures and Properties

In nature, TiO$_2$ usually has three different crystal structures: anatase, rutile, and brookite [48]. In addition, there are several metastable crystal structures of TiO$_2$ such as TiO$_2$ (H) and TiO$_2$II [49]. These metastable crystal structures can be obtained by artificial synthesis. Rutile is the most stable crystal form of TiO$_2$. Even when the particle size is reduced to the nanometer level, rutile is still the most stable TiO$_2$ nanomaterial. Anatase and brookite can be transformed into rutile at high temperature. Different crystal types of TiO$_2$ usually exhibit different morphologies and properties. Therefore, the synthesis methods and conditions for different crystal types of TiO$_2$ nanomaterials are also different. For example, the synthesis of anatase TiO$_2$ nanomaterials usually requires solution synthesis or low temperature chemical vapor deposition, however, the synthesis of rutile TiO$_2$ nanomaterials requires high temperature deposition or heating reaction [50].

Figure 1 is the crystal structure of three different TiO$_2$ phases, and the differences in crystal structures are quite evident. Rutile TiO$_2$ has a tetragonal structure (Figure 1b), and its {011} and {100} crystal facets have the lowest energy, therefore, its thermodynamically stable morphology is a truncated octahedron. Anatase has a tetragonal structure, and its c-axis is longer than the a-axis (Figure 1a). Anatase TiO$_2$ also has a low energy crystal plane, which is the same as rutile, so it can show as a truncated octahedron. The brookite belongs to an orthorhombic structure, and its structural unit is relatively larger, which is composed of eight TiO$_2$ units (Figure 1c).

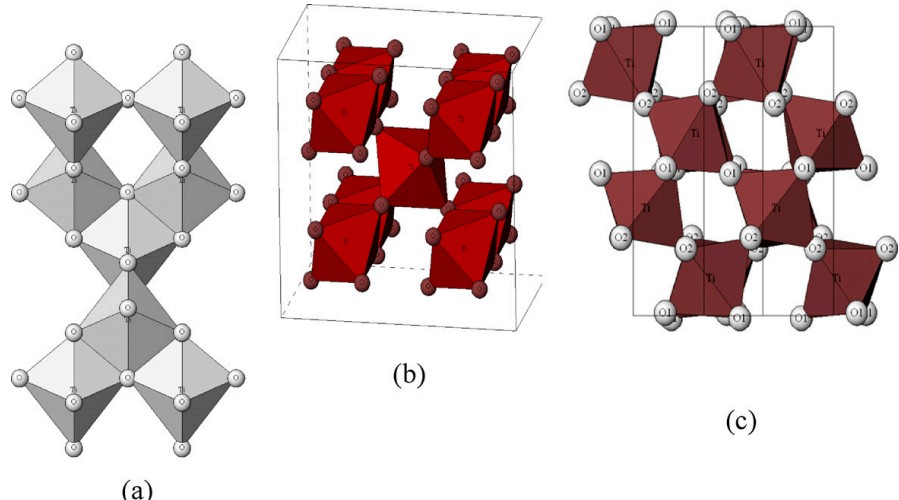

(b)

(c)

(a)

**Figure 1.** Crystalline structures of titanium dioxide (**a**) anatase, (**b**) rutile, and (**c**) brookite. Reprinted with permission from Advanced Industrial Science and Technology (AIST) https://staff.aist.go.jp/nomura-k/english/itscgallary-e.htm.

Jinfeng Zhang et al. [51] calculated the electronic structure and the effective mass of the carrier for anatase, rutile, and brookite TiO$_2$ by using the plane-wave pseudopotential method, to prove that anatase has higher photocatalytic performance than rutile and brookite. The calculation results showed that rutile had the narrowest band gap of 1.86 eV, and the band gaps of anatase and brookite were 2.13 and 2.38 eV, respectively. However, anatase is an indirect band gap semiconductor, and rutile and

brookite both belong to the direct band gap semiconductor. Therefore, this leads to longer lifetimes of photogenerated electrons and holes for anatase than those for rutile and brookite. The valence bands of $TiO_2$ in anatase, rutile, and brookite are mainly composed of O 2p and mixed with a few Ti 3d. Above the Fermi level, the conduction band is composed of Ti 3d, mixed with a small amount of O 2p and Ti 3p. The calculation results show that the anatase has a smaller average effective mass of photogenerated electrons and holes than rutile and brookite. The smaller the effective mass of the photogenerated electrons and holes, the easier it is for them to migrate, thus improving the photocatalytic activity. As anatase has a smaller effective mass and a longer lifetime of photogenerated electrons and holes, in general, anatase $TiO_2$ has a higher photocatalytic activity.

The energy level structure of semiconductor material contains two aspects: energy level position and energy band width. The position of its energy level determines whether the photocatalytic reaction can take place, and the energy band width determines its light absorption range. The position of the titanium dioxide energy level is decisive for the photocatalytic reaction. From a thermodynamic point of view, when the reduction potential of the reactant is lower than the conduction band of the semiconductor material, a reduction reaction can occur; whereas when the oxidation potential is higher than the valence band of the semiconductor material, an oxidation reaction can occur [52]. Taking the photolysis of water as an example, the generation of $H_2$ is the process of reducing $H^+$ by photogenerated electrons, while the generation of $O_2$ molecules is the process of oxidizing $O^{2-}$ by holes. The energy band position of $TiO_2$ is suitable for the photolysis of water, because the valence band of $TiO_2$ (+2.7 V, pH = 7) is lower than the redox potential of $O_2/H_2O$ (+1.23 V, pH = 7) and the position of its conduction band (−0.5 V, pH = 7) is higher than the position of $H_2O/H_2$ redox potential (−0.41 V, pH = 7) [53].

In addition to the position of the energy level, the band width also has a very important effect on photocatalytic performance. For example, the band width (3.2 eV) of $TiO_2$ is wide, so it can only absorb ultraviolet light. It is only possible to use visible light when $TiO_2$ is doped with some metals, non-metallic elements, or combined with other semiconductors with smaller energy band widths. For example, g–$C_3N_4$ has a moderate forbidden band width (2.7 eV), and its conduction band position is very high (−1.3 V, pH = 7) [54]. Therefore, the combination of $TiO_2$ and g–$C_3N_4$ can improve the utilization of visible light.

*2.2. Nano-$TiO_2$ Morphology*

A large number of studies have shown that the morphology of nanomaterials has a very important effect on photocatalytic performance, because the morphology usually determines the exposure of the crystal plane and active site, specific surface area, electron, and hole transport rate and other factors.

Zero-dimensional $TiO_2$ nanomaterial has an isotropic structure and can expose all crystal planes (including those with higher energy), which is conducive to photocatalytic reactions. However, due to the quantum confinement effect, it has a larger forbidden band width and more surface defect states, making the photogenerated electrons and holes to have a higher recombination efficiency. If the surface can be properly modified, the recombination efficiency of electron-hole pairs can be greatly reduced, which is conducive to improving the photocatalytic performance of zero-dimensional $TiO_2$ nanomaterials. This improved method has also been applied to many other zero-dimensional semiconductor nanomaterials such as carbon dots [55,56], CdS [57], CdSe [58], and graphene quantum dots [59,60].

The one-dimensional structure of $TiO_2$ such as nanorods, nanowires, and nanotubes possesses a very fast charge transfer rate in a single direction, and the electron-hole pair has a relatively low recombination efficiency, making it an important research object for photocatalytic reactions [61,62].

Two-dimensional $TiO_2$ nanosheets are very thin, with large specific surface area and effective absorption area, and the rate of charge transfer is also very fast. Therefore, two-dimensional sheet $TiO_2$ material is also widely used in photocatalysis [63,64].

In recent years, hierarchical structure $TiO_2$ nanomaterials composed of multiple morphologies have also been used in photocatalytic reactions [65,66]. These hierarchical structures of $TiO_2$

can simultaneously combine the advantages of different structures and effectively improve their photocatalytic performance.

Macak et al. [67] successfully prepared idealized $TiO_2$ nanotubes by anodizing a Ti substrate with a glycol electrolyte containing $NH_4F$ and exploring the oxidation conditions (Figure 2a–c). The presence of hexagonal nanotubes can be clearly seen from the entire layer, arranged in neat rows, with each nanotube remaining hexagonal from top to bottom. The lower wall thickness was about 65 nm and the upper wall thickness was about 12 nm. The diameter of the internal pipe increased gradually from about 50 nm to 110 nm.

Fang et al. [68] reported a new synthesis method for $TiO_2$ nanometer flowers with a large amount of {001} crystal surface exposed (Figure 2d). These nanometer flowers were completely assembled from $TiO_2$ nanometer flakes with a size of about 2.0 μm, with a thickness of about 10–20 nm and a length of about 1.2 μm.

The submicron scale hollow sphere of $TiO_2$ not only has a large specific surface, but also has a size near the wavelength of UV–Vis. Therefore, in theory, diffraction and reflection caused by shell structure on the hollow sphere can improve the utilization rate of light [69]. In the presence of cationic polystyrene sphere (PS) templates, Yoshihiko Kondo et al. [70] prepared submicron hollow sphere $TiO_2$ by hydrolyzing isopropyl titanate (Figure 2e). Uniform anatase $TiO_2$ hollow pellets with a diameter of about 490 nm and a shell thickness of about 30 nm were obtained. The resulting surface area measured by Brunauer–Emmett–Teller was 70 $m^2$/g. The photocatalytic properties were tested by the decomposition of isopropanol under ultraviolet light.

Shuai Chen et al. [71] prepared anatase $TiO_2$ nanorods by electrospinning and roasting. As shown in Figure 2f, the nanorods were observed to be 200 nm to 2 μm in length and 60 nm to 150 nm in diameter. The electrical properties of $TiO_2$ nanobelts on curved surfaces with different curvature and their photoelectric properties under different light intensities were studied. The results showed that $TiO_2$ nanobelts have potential applications in flexible photodetectors and solar cells.

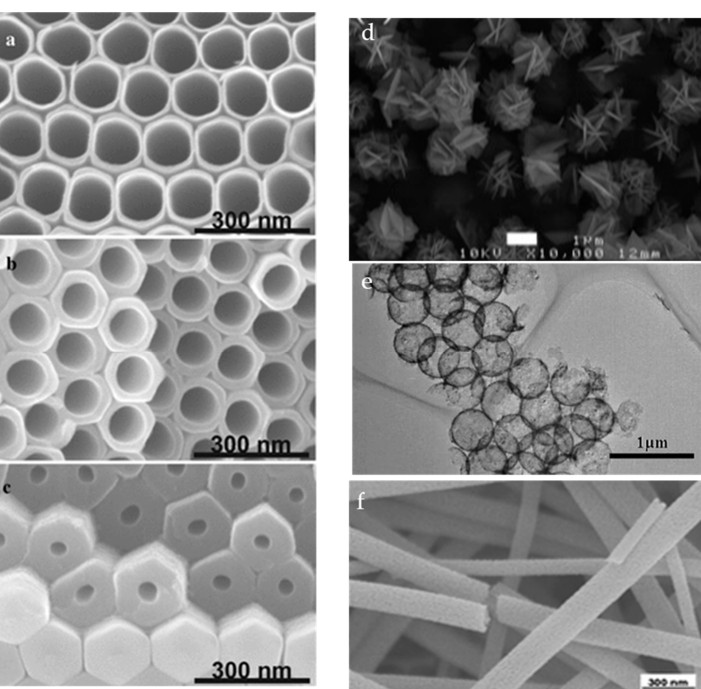

**Figure 2.** Scanning Electron Microscope(SEM) images of $TiO_2$ nanotubes taken from the upper part of the layer (**a**), the middle of the layer (**b**), and the bottom of the layer (**c**), reprinted with permission from [70]. (**d**) SEM images of flower $TiO_2$ reprinted with permission from [72], (**e**) Transmission Electron Microscope (TEM) image of $TiO_2$ hollow spheres reprinted with permission from [68]. (**f**) SEM images of $TiO_2$ nanobelts reprinted with permission from [70].

*2.3. Strategies for Improving TiO₂ Photoactivity*

Photogenic carrier recombination is a major barrier to limiting photocatalytic activity of semiconductors [71]. When recombination occurs, the excited electrons return to the valence band [73], which then do not participate in the reaction and dissipate energy in the form of radiation [74,75]. Recombination can occur on a surface as a whole, and the introduction of impurities or crystal defects can affect recombination [76]. It has been reported that doping ions [77–79], heterojunction coupling [80–82], and nanometer crystals [83,84] can promote electron-hole pair separation and reduce recombination, thus increasing photocatalytic activity.

2.3.1. Metal Doping

The absorption of visible light by wide-band gap semiconductor was originally realized by doping metal elements. According to the semiconductor band theory, due to the difference in the valence state between doped metal elements and metal elements in the semiconductor, the doped metal elements can generate donor or acceptor levels in the band gap of the semiconductor. The donor (or acceptor) energy level has two states, deep or shallow, due to the strength of the energy level binding to the electron.

As shown in Figure 3, the shallow donor level exists below the semiconductor conduction band (Figure 3a) and the shallow acceptor level exists above the semiconductor valence band (Figure 3b), while the deep donor level is close to the valence band in the semiconductor band (Figure 3c), and the deep acceptor level is close to the conduction band in the semiconductor band (Figure 3d). Electrons will jump between the donor level (or valence band) and the conduction band (or acceptor level), where the transition from the shallow donor level to the conduction band (or from the valence band to the shallow acceptor level) is a shallow transition, and the transition from the deep donor level to the conduction band (or from the valence band to the deep acceptor level) is a deep transition. Since the energy barrier to be crossed for shallow or deep transitions is smaller than the intrinsic band gap of the semiconductor, visible light can excite shallow transitions and, in most cases, deep transitions.

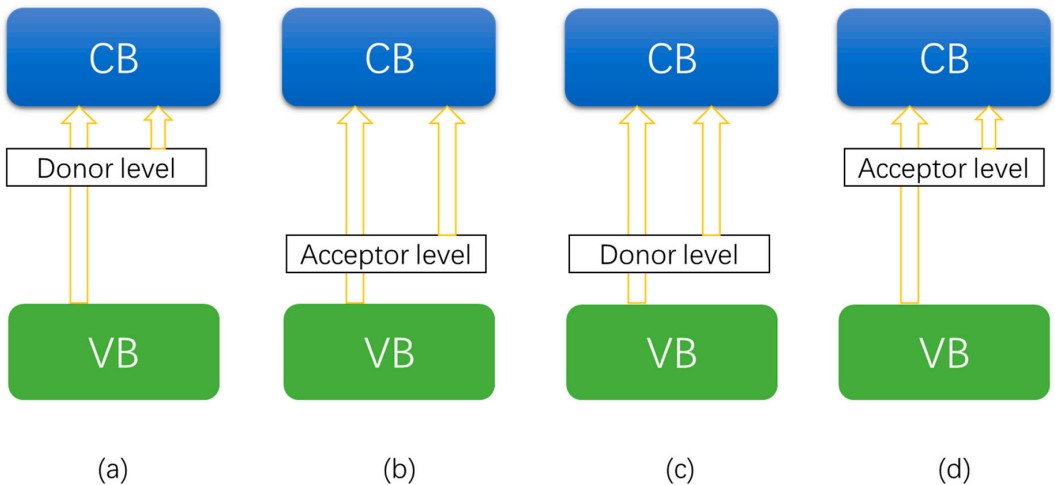

**Figure 3.** Effects of metal doping on band structure of semiconductors: (**a**) shallow donor level, (**b**) shallow acceptor level, (**c**) deep donor level, and (**d**) deep acceptor level formed by metal doping.

Metal doping can indeed introduce impurity levels into the wideband gap semiconductor band, and these impurity levels can also induce the absorption of visible light, but not always the improvement of photocatalytic efficiency under visible light. The current widely accepted explanation for this problem is that metal dopants may become the composite centers of photogenerated electrons and holes (especially the formed deep impurity level) in semiconductor materials, thus failing to improve (or even reduce) the visible light activity of semiconductor materials. Moreover, the visible light activity of semiconductor materials is affected by the type of doping elements, the doping method (chemical

synthesis, atmospheric heat treatment, magnetron sputtering, etc.), the doping amount, the doping position (substitution or clearance), and the doping distribution (volume or surface).

### 2.3.2. Non-Metal Doping

Due to the bottleneck problem of metal doping, non-metal doping research has gradually become the mainstream since 2001, when Asahi et al. [85] reported the visible activity of N-doped $TiO_2$. In the past decade, important progress has been made in studies on non-metal doping including synthesis, characterization, mechanism research, and performance.

According to the band theory of semiconductors, the energy band of the semiconductor is a new molecular orbital formed by the hybridization of atomic orbitals of constituent atoms. The conduction band and valence band of most transition metal oxides (including $TiO_2$) are mainly composed of 3d metal orbital and 2p oxygen orbital, respectively. Again, on the basis of molecular orbital theory, when some of the oxygen atoms in the lattice of $TiO_2$ are replaced with an element that is less electronegative than oxygen, the electron orbitals of the doping elements with the 2p orbital of oxygen forms a new molecular orbital with lower energy than the 2p orbital of oxygen, that is, the top of valence band formed by this new molecular orbital is higher than the top of the valence band formed by the 2p orbital of oxygen. However, the 3d orbit of the titanium forming the conduction band does not change, so it can be seen that through the doping of elements with lower electronegativity than oxygen, the bottom of the conduction band for titanium dioxide remains unchanged, and the top of the valence band is increased, thus reducing the band gap of $TiO_2$. In fact, most non-metallic elements are less electronegative than oxygen, which can be used to reduce the band gap of $TiO_2$ through doping. In addition, a suitable dopant element, in addition to the requirement of electronegativity, should have an ion radius similar to that of oxygen ions in order to achieve atomic substitution doping.

### 2.3.3. Vacancy

Heteroatomic doping of semiconductors also typically introduces vacancies in the lattice. Vacancy is one of the defects of intrinsic characteristics in semiconductor materials such as oxygen vacancy in oxides and nitrogen vacancy in nitrides. Vacancy can introduce a defect level into the band gap of the semiconductor (for example, the defect level of the oxygen vacancy is located below the conduction band), thus causing visible light absorption of the semiconductor material. There are many studies on vacancy (especially oxygen vacancy [86,87]), and in most cases, the presence of vacancy can improve the catalytic activity of semiconductor materials. For example, the introduction of nitrogen vacancy in $g-C_3N_4$ can effectively improve the photodissociation performance of aquatic hydrogen [88]. In addition, the presence of defect levels can even enable the insulator $SiO_2$ (band gap > 8 eV) to have the capability of photocatalytic hydrogen production [89].

### 2.3.4. Composites

Photoelectron and hole in semiconductor materials are mostly recombined in the process of bulk phase diffusion or transferred to the surface, and only a few electrons and holes participate in redox reaction, which is the most fundamental factor that limits the photocatalytic activity of semiconductor materials. For a single material, a defect on its surface (such as a vacancy, etc.) will become the trapping pit of an electron or hole, thus causing the separation of photogenerated electrons and holes, but the separation is finite. To obtain more separated electrons and holes, researchers have used carrier transfers between composite semiconductors over the past few decades.

Figure 4a shows the carrier transfer in the semiconductor sensitization process, which mainly expands the light absorption range of composite semiconductor materials. Generally, the semiconductor with narrow band gap and high conduction band is combined with a semiconductor with a wide band gap and low conduction band (such as $TiO_2$, ZnO, etc.). In this kind of composite semiconductor material, the electrons, which can be excited by visible light to the conduction band of the narrow band

gap semiconductor, are transferred to the conduction band of a wideband gap semiconductor under the drive of energy level difference.

If two semiconductor materials with staggered energy levels are combined to form a type II semiconductor heterostructure, the carrier transfer between them is shown in Figure 4b. Electrons transfer from a semiconductor with a high conduction band to a semiconductor with a low conduction band under the action of energy level difference, while holes transfer from a semiconductor with a low valence band to a semiconductor with a high valence band, thus facilitating the spatial separation of photogenerated electrons and holes. So far, most semiconductor materials can find another semiconductor material matching its energy level, and form this type II semiconductor heterostructure, and type II energy level arrangement is also the most commonly used in semiconductor combination of the semiconductor heterostructure. For example, $ZnS/ZnO$ [90], $SnO_2/ZnO$ [91], $TiO_2/WO_3$ [92], $g–C_3N_4/WO_3$ [93], etc.

If the semiconductor materials that make up the type II semiconductor heterostructure are p-type and n-type, respectively, and their band edges are arranged as shown in Figure 4c, then the built-in electric field in the p-n junction will further increase the carrier transfer to promote greater separation of photogenerated carriers on semiconductors. Such semiconductor heterostructures are $p-In_2O_3/n-ZnO$ [94], $p-CaFe_2O_4/n-ZnO$ [95], $p-NiO/n-ZnO$ [96], and $p-ZnO/n-TiO_2$ [97], etc.

The above-mentioned type II semiconductor heterostructure can promote the separation of photogenerated carriers of semiconductors, but there are also unfavorable factors, that is, the carriers are all transferred to the low energy level, thereby the reduction and oxidation capacity of photogenerated electrons and holes is reduced. If the separation of photogenerated carriers can be achieved while maintaining their ability for oxidation and reduction, it will be beneficial for the application of semiconductor heterostructures in photocatalysis. For this, the research in some specific II type semiconductor heterostructures (for example, the $CdS/ZnO$ [98], $WO_3/CaFe_2O_4$ [99], $CdS/WO_3$ [100], $BiOI/g–C_3N_4$ [101], etc.) puts forward the concept of the mechanism of the vector Z carrier transfer (Figure 4d). Taking $CdS/ZnO$ as an example, during the carrier transfer process of the vectorial Z-scheme mechanism, the photogenerated electrons on ZnO and the photogenerated holes on CdS recombine at the interface, thus, the photogenerated electrons on CdS with stronger reducibility and the photo-generated holes on ZnO with stronger oxidability are retained.

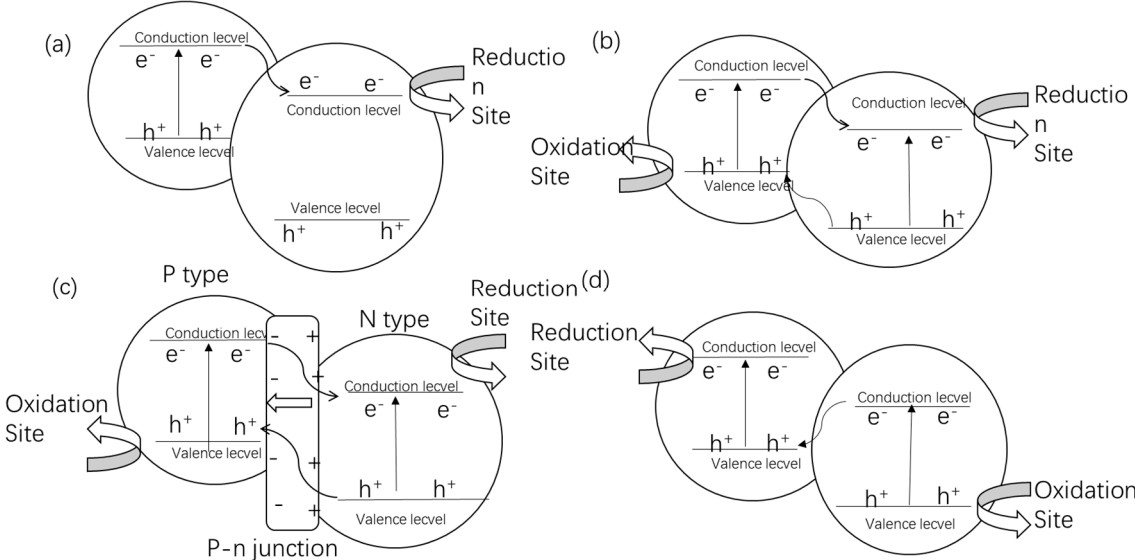

**Figure 4.** Charge-carrier transfer process in sensitization (**a**), traditional (**b**), p-n junction (**c**), and vectorial Z-scheme (**d**) mechanisms.

### 3. Photocatalytic Hydrogen Production

Hydrogen (H$_2$), with its heat of combustion, is expected to be an important energy source on the world energy stage. So far, a number of methods have been implemented to obtain hydrogen such as chemical, physical, or electrocatalytic hydrogen production [102–104]. In these methods, photocatalytic hydrogen production [105,106] has developed rapidly because it is clean, cheap, and environmentally friendly. Many materials with excellent photocatalytic hydrogen production properties have been reported such as ZnO [107], MoS$_2$ [108], etc. In particular, TiO$_2$ [109,110], which has physical and chemical stability, a unique energy band structure and photochemical activity, was first reported in 1972 [23], and has since been a research hot spot.

The principle of photocatalytic hydrogen production is shown in Figure 5 [111,112]. Photocatalysis has four main processes in Figure 5, namely, light collection (stage 1), the electron excites from the valence band to the conduction band (stage 2), photogenerated electrons and holes separation and transfer (stages 3 and 4), and surface redox reactions (stages 5 and 6) [54]. First, the energy required for photocatalysis needs to be above or equal to the band gap of the semiconductor. Generally, the semiconductor consists of the valence band and the conduction band, which are separated from each other by the forbidden band [113]. Under the appropriate conditions, the photocatalyst excites to produce electron and hole pairs (e$^-$ + h$^+$) (Equation (1)), the electrons are excited from the valence band to the conduction band, leaving holes in the valence band. The recombination of photoelectrons and photoholes also occurs as Equation (2). Photogenerated electrons and holes participate in the redox reactions in water. The oxidation reaction is that the water reacts with h$^+$ to produce O$_2$, as shown in Equation (3), while Equation (4) shows the reduction reaction is that H$^+$ gains e$^+$ to produce H$_2$. When reduction and oxidation occur, the redox reaction potentials on the photocatalyst surface are higher than the conduction band and lower than the valence band levels [54,114,115]. Photogenerated holes have strong oxidation ability to oxidize water and organics (such as alcohols), as shown in Equations (3) and (5). The thermal dissociation of water can be carried out at a temperature higher than 2070 K, but the photocatalyst can be used under the light radiation to decompose water at room temperature with an energy greater than the band gap energy [116]. The mechanism of photocatalytic dye degradation [117], carbon dioxide reduction [118], and nitrogen fixation [119] is similar to the mechanism of photocatalytic hydrogen production, except that the redox potential of the reaction is different, and is not repeated below.

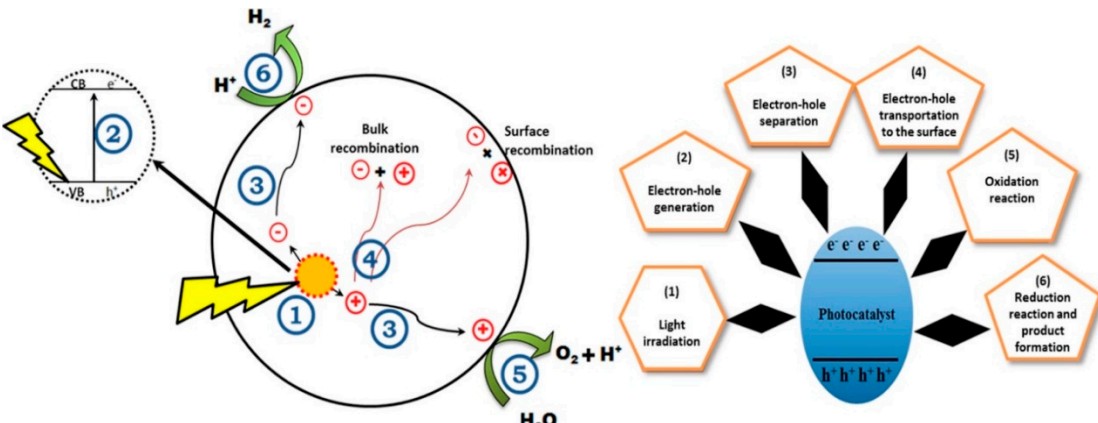

**Figure 5.** Photocatalytic hydrogen production mechanism. Reprinted with permission from [111].

$$\text{Catalyst} \rightarrow \text{Catalyst (e}^- + \text{h}^+) \tag{1}$$

$$\text{Catalyst (e}^- + \text{h}^+) \rightarrow \text{Catalyst} \tag{2}$$

$$H_2O + h^+ \rightarrow \frac{1}{2}O_2 + 2H^+ \tag{3}$$

$$2e^- + 2H^+ \rightarrow H_2 \tag{4}$$

$$RCH_2OH + 2h^+ \rightarrow RCHO + 2H^+ \tag{5}$$

The photocatalytic ability of rutile TiO$_2$ is in principle superior to anatase [120]. First, rutile is thermodynamically stable, while anatase is metastable [121,122]. Second, rutile exhibits more effective charge separation due to fewer defects in its crystal, because the defect is the recombination center of the photogenerated electron-hole pair [123]. Most importantly, rutile titanium dioxide has a smaller band gap (3.02 eV) than anatase (3.20 eV) [124–126]. Therefore, in theory, rutile TiO$_2$ has greater potential as a photocatalyst. However, due to the high photogenerated electrons and holes recombination rate of rutile TiO$_2$, it usually shows a lower photocatalytic performance than anatase TiO$_2$ [127–129]. Inspired by the concept of "surface heterojunction" [130], Chaomin Gao et al. [131] introduced the facet heterojunction (FH) strategy to promote the separation of photogenerated electrons and holes in rutile TiO$_2$ by building 3D layered nanostructures to obtain higher photocatalytic activity. A 3D rutile TiO$_2$ photocatalyst was prepared on the surface of TiO$_2$ nanorods (NRs) coated with ultrathin TiO$_2$ nanocrystalline sheets. This 3D rutile TiO$_2$ contained countless FH, which are formed by nanosheets with different levels on the interface between the coated nanosheets and the nanorod substrate. In order to clarify its high activity, FH-TiO$_2$ was compared with rutile TiO$_2$, anatase TiO$_2$, and P25, and the yield of H$_2$ production for FH-TiO$_2$ was 1.441 mmol g$^{-1}$ h$^{-1}$, which was much higher than the other photocatalysts (Figure 6).

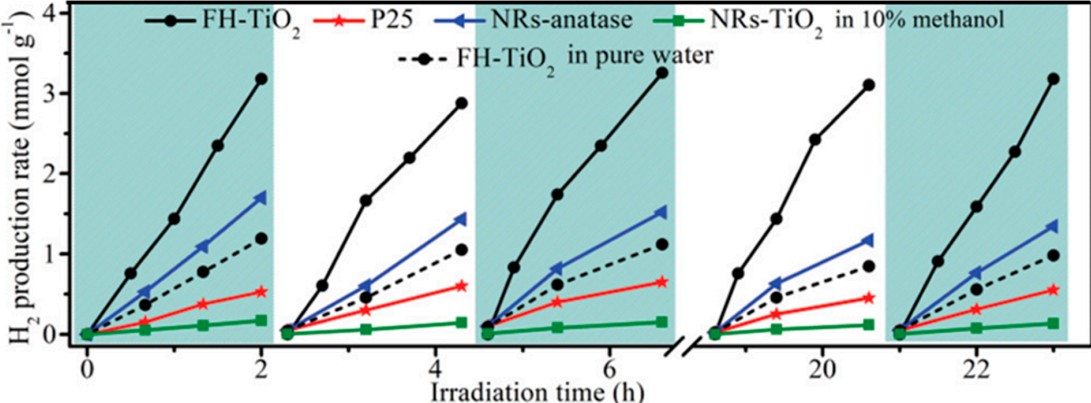

**Figure 6.** Photocatalytic hydrogen production cycle experiment. One wt% Pt as a cocatalyst was loaded on FH-TiO$_2$, P25, NRs-anatase, and NRs-TiO$_2$. Reprinted with permission from [131].

A large number of mesoporous TiO$_2$ materials with interpenetrating and regular mesoporous systems have been studied in a large body of research [132–134]. They have great potential in photocatalysis [135,136]. At present, many efforts have been made to pursue high-performance mesoporous TiO$_2$ (OMT) materials, but the results have not been good [137–140]. Wei Zhou et al. [141] (Figure 7) synthesized mesoporous black TiO$_2$ (OMBT) with regular pore sizes and large specific surface area (124 m$^2$ g$^{-1}$). OMBT materials can extend the optical response to visible and even infrared regions. The photocatalytic hydrogen production activity of OMBT was very high (136.2 μmol h$^{-1}$), and there was no significant inactivation after multiple cycles. The band gap of OMBT was significantly smaller than that of OMT, making it more susceptible to visible light excitation.

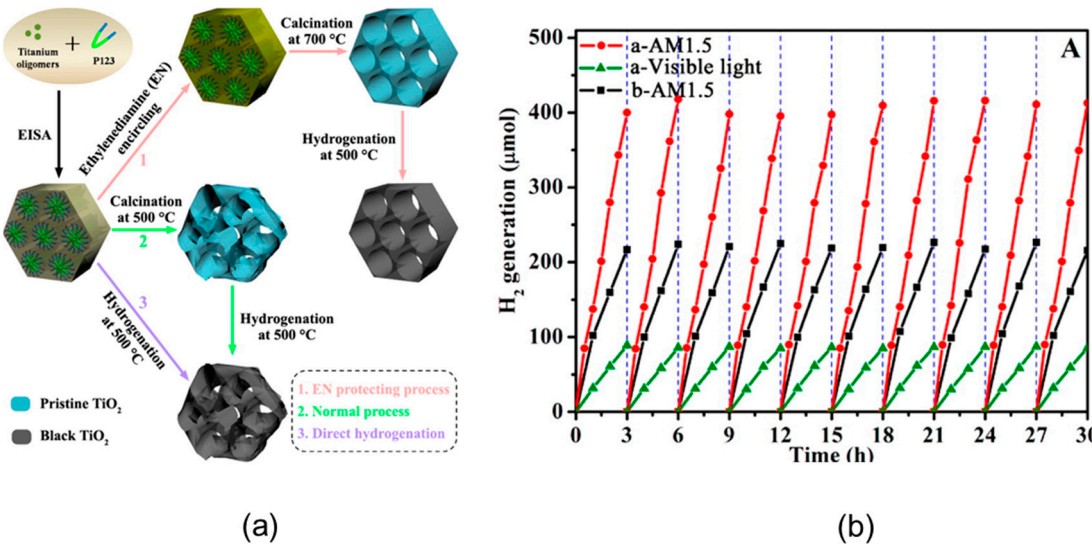

(a)                                                                                     (b)

**Figure 7.** (**a**) The synthesis process of mesoporous black TiO₂(OMBT). (**b**) Photocatalytic hydrogen production cycle experiment under AM 1.5 and visible light irradiation. Reprinted with permission from [141].

The exposed surface of anatase TiO₂ also determines the photocatalytic capacity of water to a large extent. In order to obtain higher photocatalytic activity, shape control strategy has always been a hot research topic [102,142]. In recent years, it has been theoretically revealed that the {001} and {101} faces can effectively separate electrons and holes, so the photocatalytic activity can be improved [143]. The {101} face (0.44 J m$^{-2}$) has a lower surface energy than the {001} face (0.90 J m$^{-2}$), so the {101} face is more thermodynamic stable [144]. Various reaction systems have been developed to control the growth of anatase TiO₂ to increase the percentage of {001} face [145]. Ming Li et al. [146] (Figure 8) used ethanol as the reaction solvent and regulated the growth of anatase TiO₂ to obtain TiO₂ with high {001} surface exposure, and added F$^-$ to further reduce the surface energy of TiO₂. Through this method, the thickness of anatase TiO₂ nanosheets can be successfully reduced to ≈2.5 nm, side length ≈200 nm, and {001} plane accounting for 97%. Using Pt as a cocatalyst, TiO₂ nanosheets showed high activity (17.86 mmol h$^{-1}$ g$^{-1}$) at UV (365 ± 10 nm) with an apparent quantum efficiency (QE) of 34.2%.

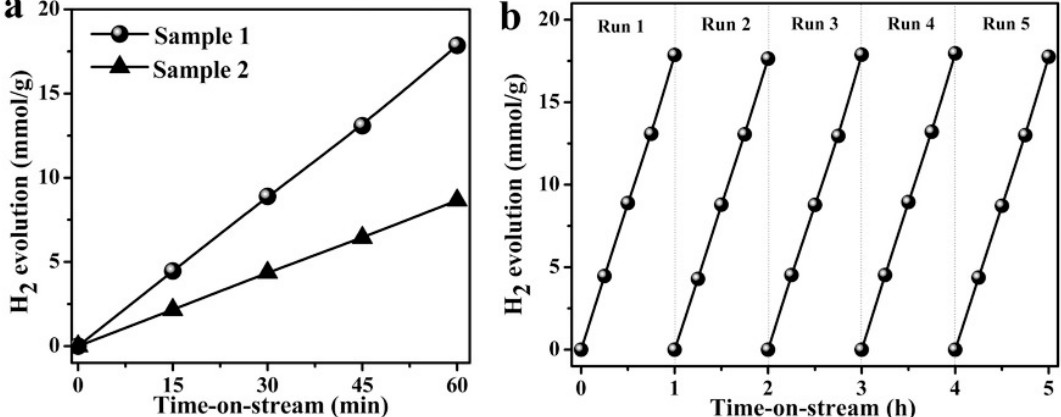

**Figure 8.** Ethanol was added during the synthesis of sample 1 and no ethanol in sample 2. (**a**) Photocatalytic hydrogen production of anatase TiO₂ nanosheets under UV, (**b**) Photocatalytic hydrogen production cycle experiment of sample 1 under UV. Reprinted with permission from [146].

Loading the oxidation or reduction cocatalyst on semiconductor surfaces is a broadly used and effective strategy, which can separate effectively photogenerated electrons and holes and generate

surface redox reaction sites [147–150]. Although theoretically loading the oxidation or reduction cocatalyst simultaneously can enhance photocatalytic activity, in most cases, the cocatalyst will be randomly distributed on the surface of the semiconductor, which leads to a higher recombination rate of photogenerated electrons and holes [151]. Due to the limitations of traditional methods in preparing this type of photocatalyst, there have been few reports of success. Atomic layer deposition (ALD) is a new and effective method for the preparation of high dispersion loaded materials [152]. Jiankang Zhang et al. [153] (Figure 9) used the ALD method to modify Pt internally and $CoO_x$ externally in surfaces of porous $TiO_2$ nanotubes with carbon nanotubes as template, and synthesized a new porous tubular $CoO_x/TiO_2/Pt$ photocatalyst, which was successfully used in photocatalytic hydrogen production. The photogenerated electrons and holes flow inward and outward, respectively, in the porous titanium dioxide nanotubes. It can be seen that the highest activity was found when both Pt and $CoO_x$ were loaded at the same time, which was higher than that when Pt and $CoO_x$ were loaded alone.

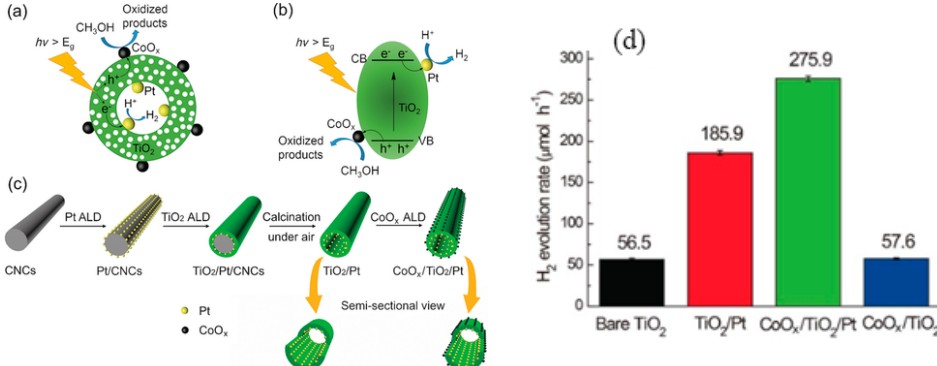

**Figure 9.** (**a**,**b**) The mechanism of photocatalytic hydrogen production by CoOx/TiO$_2$/Pt. (**c**) Synthetic process of $CoO_x/TiO_2/Pt$ by atomic layer deposition (ALD). (**d**) Photocatalytic hydrogen production of different photocatalysts in a 15 vol% methanol–water solution under UV. Reprinted with permission from [153].

In addition to Pt, some other noble metals are used as cocatalysts or dopants to enhance the photoactivity of the semiconductor. Alenzi et al. [154] synthesized a $TiO_2$ nanofilm doped with Ag, and this composite film showed a greater hydrogen production activity with an average hydrogen production rate of $147.9 \pm 35.5$ µmol h$^{-1}$ g$^{-1}$ than $TiO_2$. Using calculations based on density functional theory (DFT), Mazheika et al. [155] found that Ag could modify the surface of anatase $TiO_2$, resulting in defects such as oxygen vacancy when Ag was introduced into the $TiO_2$ lattice [156]. With these vacancies, a transmission channel with high conductivity can be created, thereby effectively separating electrons from holes [157]. Yingfeng Xu et al. [158] developed a particularly simple and effective palladium catalytic strategy for hydrogenation of $TiO_2$. The point defects ($Ti^{3+}$ and oxygen vacancy) in $TiO_2$ significantly narrow the band gap and improve the activity of $TiO_2$. In addition, due to the oxygen assistance, the autothermal effect generated during palladium-catalyzed hydrogenation induces a unique crystalline core/disordered shell structure, which is believed to prevent photogenerated electors and holes from recombination and is conducive to the cycling stability of $TiO_2$ after redox reduction. This simple and universal palladium catalytic hydrogenation method will open up new ways to produce inherent defects in oxides and greatly improve catalytic performance.

Non-metallic element doped $TiO_2$ has obvious advantages due to its unique characteristics of small ionic radius, high thermal stability, and few recombination centers. Studies have shown that N or S [159–161] doped $TiO_2$ can narrow the optical band gap and increase photocatalytic activity. Recently, a series of N-doped $TiO_2$ nanobelts have been reported by Shuchao Sun et al. [162]. The unique shape of the nanobelt provides well-defined nanostructures with (101) and (001) faces on exposed surfaces. The maximum hydrogen yield rate of these N-doped $TiO_2$ nanoribbon is 670 µmolh$^{-1}$g$^{-1}$, much higher than other values reported in the previous $TiO_2$ nanobelt conventional literature (only a few µmol h$^{-1}$g$^{-1}$).

S doped $TiO_2$ was prepared by oxidative annealing of $TiS_2$ as reported by Umebayashi et al. [163]. At the annealing temperature of 600 °C, a part of $TiS_2$ was transformed to anatase $TiO_2$. The residual S atoms formed S-doped $TiO_2$ by the Ti–S bond. The S-doped energy band structure is calculated by the super-cell method. It was found that when $TiO_2$ mixed with S 3p increased the height of the valence band, it led to narrowing of the band gap. As the narrowing of the band gap is caused by the upward migration of valence bands, the conduction band remains unchanged, so S-doped $TiO_2$ can produce hydrogen in visible light.

In photocatalysis, $TiO_2$ can be combined with other semiconductors to extend the absorption wavelength range to the visible region and slow down electron-hole recombination [164,165].

Kim et al. [166] reported a new system using a $TiO_2$ photocatalyst loaded with $Bi_2S_3$ ($Bi_2S_3/TiO_2$). $Bi_2S_3$ particles are similar to the shape of a sea urchin, with a length of about 2–3 nm and a diameter of 15–20 nm. Compared with pure $TiO_2$ and $Bi_2S_3$, $Bi_2S_3/TiO_2$ composite material has enhanced hydrogen generation capacity in methanol/water (1:1) system.

Chai et al. [167] reported g–$C_3N_4$–Pt–$TiO_2$ nanocomposite was prepared by simple chemical adsorption and calcination. Through a series of characterization, it was found that the photocatalytic activity of the composites could be significantly improved. The photocurrent stability of g–$C_3N_4$–$TiO_2$ is about 1.5 times that of g–$C_3N_4$. Due to the synergistic effect between Pt–$TiO_2$ and g–$C_3N_4$, the photogenerated electrons of g–$C_3N_4$ can transform to Pt–$TiO_2$, and the photogenerated electrons and holes can be effectively separated, which can effectively hinder the rapid recombination of photogenerated electrons and holes and improve the photocatalytic activity.

Some typical examples of improving photocatalytic activity are described above, which have given us great inspiration. In addition, Table 1 lists the basic methods commonly used in the literature to improve photocatalytic activity such as metal doping, non-metal doping, and composites. Through comparison, it can be seen that in the process of photocatalytic hydrogen production, due to the advantages of the nanosheet structure, most materials were synthesized into thinner nanosheets in order to seek rapid electron migration to the surface and therefore improve catalytic activity. The highest activities were Co (III)/$TiO_2$ and F–$TiO_2$, showing that metal and non-metal doping could significantly improve the photocatalytic activity. Doping is the simplest and easiest way to improve photocatalytic activity. In the process of composites, two or three semiconductors with suitable band structures are usually combined so that the photogenerated electrons can be separated effectively. However, due to the complexity of the implementation method, it is not conducive to large-scale industrial production.

**Table 1.** Summary of common methods to improve the photocatalytic hydrogen production activity of $TiO_2$.

| Catalyst | Morphology | Reaction Conditions | Activity | Reference |
|---|---|---|---|---|
| S–$TiO_2$ | Nanopillar | AM 1.5 | 163.9 $\mu$mol h$^{-1}$ g$^{-1}$ | [168] |
| F–$TiO_2$ | Nanosheet | UV–Vis | 18,270 $\mu$mol h$^{-1}$ g$^{-1}$ | [169] |
| $TiO_2$ @$ReS_2$ | Nanorod | Visible light | 1404 $\mu$mol h$^{-1}$ g$^{-1}$ | [170] |
| P/$TiO_2$(B) | Nanofiber | AM1.5 | 380 $\mu$mol h$^{-1}$g$^{-1}$ | [171] |
| Ag/$TiO_2$ | Nanosheet | UV–Vis | 1.34 $\mu$mol cm$^{-2}$ h$^{-1}$ | [172] |
| Fe/Ni–$TiO_2$ | Nanoparticle | Visible light | 361.64 $\mu$mol h$^{-1}$ g$^{-1}$ | [173] |
| Co(III)/$TiO_2$ | Nanosheet | AM 1.5 | 20 mmol h$^{-1}$g$^{-1}$ | [174] |
| $TiO_2$:Rh/Nb | Nanorod | UV–Vis | 0.99 mmol g$^{-1}$ h$^{-1}$ | [175] |
| MoxS@$TiO_2$@$Ti_3C_2$ | Nanosheet | AM 1.5 | 10,505.8 $\mu$mol h$^{-1}$g$^{-1}$ | [176] |
| Au/$TiO_2$/SDA | Nanosheet | Visible light | 264 $\mu$mol g$^{-1}$ h$^{-1}$ | [177] |
| C-$TiO_2$/g–$C_3N_4$ | Nanosheet | Visible light | 1409 $\mu$mol h$^{-1}$ g$^{-1}$ | [178] |

## 4. Photocatalytic Dye Degradation

With the improvement in industrialization, the mass production rate of dyes has increased, so effective waste water treatment is needed [179]. Therefore, many methods have been developed to

treat this apparent challenge, but these physical techniques (e.g., activated carbon adsorption and resin adsorption, reverse osmosis, extraction, ultrafiltration) simply transfer contaminants from one medium to another, causing secondary pollution. This usually requires further treatment of the absorbent or extractant, which increases the cost of the process. In recent years, extensive studies have been carried out on a wide range of synthetic dyes. Since 1972, when $TiO_2$ was first used for photocatalytic hydrogen production, photocatalysis has attracted much attention. Since the electron with energy generated by $TiO_2$ under light degraded the dyes, a series of catalysts were developed for the photocatalytic degradation of dyes. Among the different types of photocatalysts, photocatalytic oxidation assisted by titanium dioxide ($TiO_2$) has attracted much attention in recent years due to its non-toxicity, strong oxidation capacity, and long-term optical stability [180].

The introduction of carbon nanotubes (CNT) into nano-$TiO_2$ to prepare highly active photocatalyst has attracted the attention of many researchers [181–184]. These reports indicate that when decorated with CNT, the photocatalytic activity of $TiO_2$ can be observably enhanced. It has been indicated that CNT is an electron acceptor, hinders the recombination of electrons and holes, and increases the superficial area of $TiO_2$ and the number of active sites. Azzam et al. [185] successfully synthesized the nano-composite materials by CNT, $TiO_2$, and silver nanoparticles (AgNPs) for the photodegradation of dyes. The synthesized catalyst was modified with a cationic surfactant to increase the dispersion of $TiO_2$ and reduce the surface interaction ($TiO_2$@CNT/AgNPs/C10). The results of $N_2$ gas adsorption and desorption showed that the obtained material had a large specific surface area. Due to the co-modification of Ag nanoparticles and CNT, the $TiO_2$ composite had a lower band gap (2.25 eV). It is obvious from Figure 10 that $TiO_2$ @CNT/AgNPs/C10 had the highest catalytic activity.

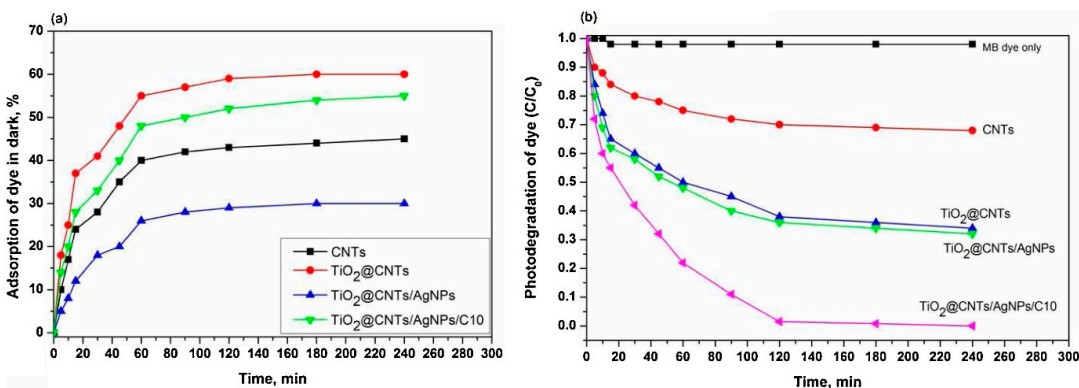

**Figure 10.** Degradation of methylene blue dye by (**a**) adsorption and (**b**) photodegradation under visible light [$C_0$ = 20 mg/L, catalyst dose = 0.5 g/L, pH = 5.8 and T = 25 °C]. Reprinted with permission from [185].

$TiO_2$ doping includes Au, Ag, Pt, and other noble metal nanoparticles (NPs) that can effectively promote the separation of photogenerated electrons and holes, and the noble metal can also act as an active site for the photocatalytic degradation of organic dyes [137,186]. In general, the smaller the size of NPs, the more active sites they provide [187]. However, the bonding properties between metals and semiconductors are largely dependent on the properties of ligands on the metal surface (organic ligands are often added to avoid NP aggregation in solution) and organic ligands on the metal surface form a physical barrier that prevents the reactants from spreading to the active site and the transfer of electrons from the nano-$TiO_2$ to the noble metal [188]. Haiguang Zhu et al. [189] synthesized a photocatalyst for $TiO_2$ nanoparticles (NPs) doped with gold nanoclusters (Au NCs), which are protected by per-6-thio-β-cyclodextrin (SH-β-SD). The use of SH-β-SD facilitates the formation of an interface between Au NCs and $TiO_2$, thus forming an effective photocatalyst and overcoming the disadvantages of organic ligands in the synthesis of traditional photocatalysts. The photocatalytic activity was evaluated by MO (Methyl Orange) degradation. As shown in Figure 11, $TiO_2$–Au NCs–β–CD exhibited higher photocatalytic activity for MO degradation under UV light than other composite materials.

In particular, over 98% of the MO was degraded using $TiO_2$–Au NCs–β–CD within 10 min, while about 47% of the MO was degraded using $TiO_2$.

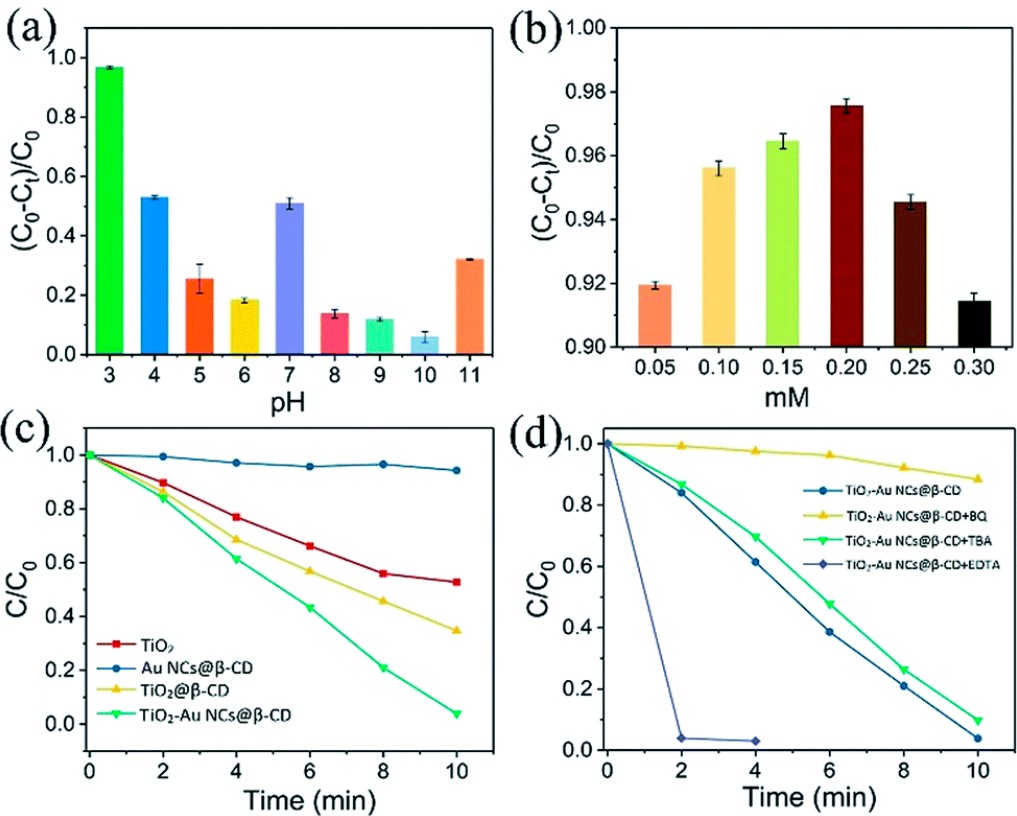

**Figure 11.** (**a**) Photodegradation efficiency of $TiO_2$–Au NCs@β-CD (t = 10 min) for MO at different solution pH. (**b**) Photodegradation efficiency of $TiO_2$–Au NCs@β-CD for MO (pH = 3), which was prepared by using different amounts of Au NCs@β-CD (t = 10 min). (**c**) Relative concentration (C/C0) versus time plot for the photodegradation of MO in the presence of various amounts of catalysts under UV light. (**d**) Relative concentration (C/C0) versus time plot for the photodegradation of MO by $TiO_2$–Au NCs@β-CD in the presence of different kinds of scavengers. Reprinted with permission from [189].

Due to the high price of noble metals and the influence of organic ligands, the photocatalytic capacity of noble metals cannot be fully developed. Recently, monatomic catalysis has displayed great potential in improving photocatalytic activity and noble metal utilization [190–192]. Monatomic metal can improve more active sites and avoid the influence of organic ligands [193]. Tongzhou Xu et al. [194] synthesized a photocatalyst for $TiO_2$ nanofilms with atom Pt injection, and $TiO_2$ nanofilms exposed a {001} face (Figure 12). Photocatalytic degradation of acetamide under vacuum ultraviolet (VUV) and ultraviolet (UV) irradiation was used to evaluate the photocatalyst activity. Photocatalytic degradation showed high activity and stability. When the initial concentration was 500 ppb and 100 ppb, respectively, the degradation rate of acetamide reached 94.52% and 100% after 5 min of irradiation, which were 2.19 and 3.98 times of the nanoporous $TiO_2$.

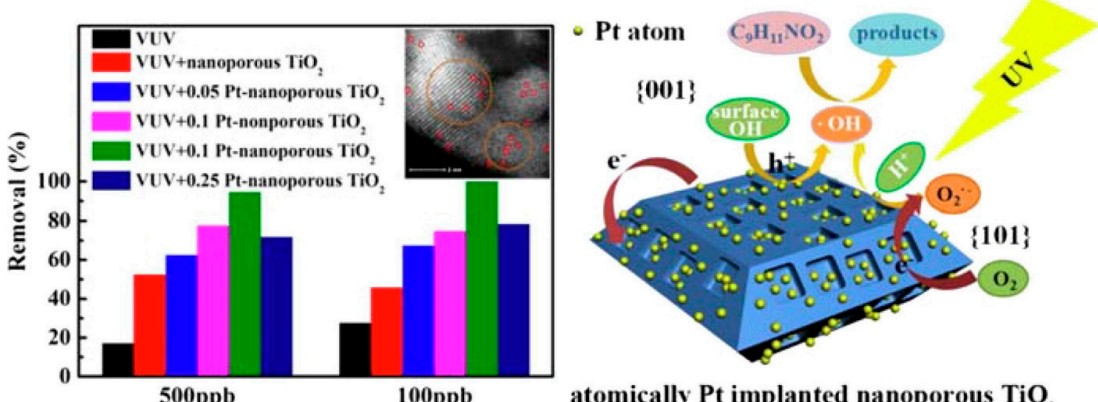

**Figure 12.** (**a**) Photocatalytic degradation of dye activity within 0.5 min (**b**) Photocatalytic mechanism atomically Pt-nanoporous TiO$_2$ with exposed {001} facets under UV irradiation. Reprinted with permission from [194].

Nano-TiO$_2$ photocatalyst powders are easily agglomerated in solution and their activity is reduced. Furthermore, dispersion in solution makes recycling difficult. These defects limit the wide application of TiO$_2$ photocatalysis in industry [195,196]. Recently, it has been reported that nano-TiO$_2$ powder fixed on the supporting material can effectively solve the problem of agglomeration and circulation [197,198]. Yang Li et al. [199] successfully prepared a photocatalyst by loading TiO$_2$ nanoparticles on polymethyl methacrylate (PMMA) nanofibers through water treatment of electrospun PMMA nanofibers containing n-butanol titanium precursors at 135 °C (Figure 13). Under the irradiation of ultraviolet lamp, 0.1 g TiO$_2$@PMMA can completely degrade 100 mL MO in 50 min, which has a high photocatalytic activity. Moreover, TiO$_2$@PMMA could be separated from the reaction liquid by filtration and remained stable in five consecutive MO degradation cycles without obvious inactivation.

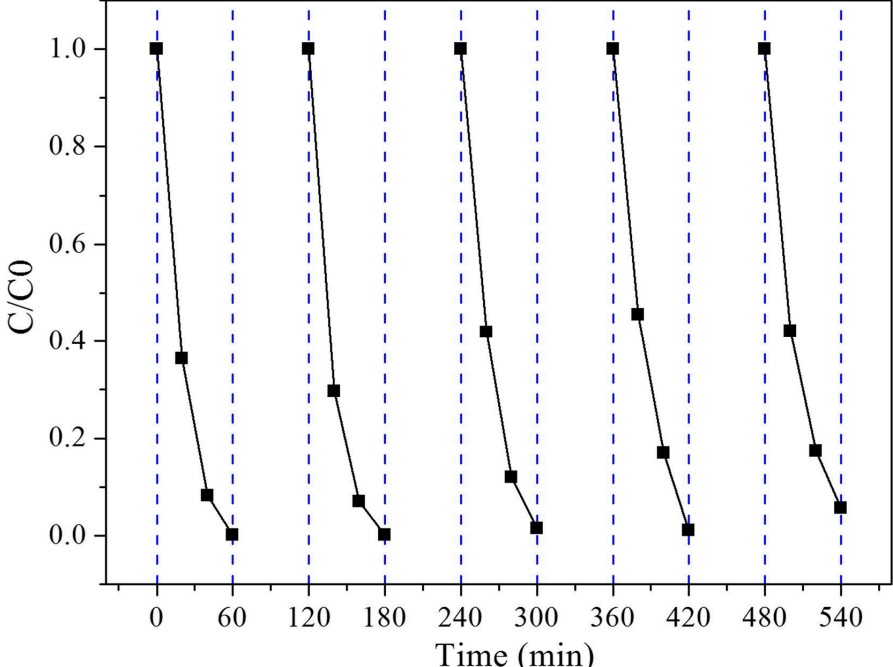

**Figure 13.** Cycle test of photocatalytic degradation of MO by TiO$_2$@PMMA ([MO] = 10 mg/L). Reprinted with permission from [199].

There have also been some studies to improve the activity of photocatalysts by simply doping $TiO_2$ with nonmetal or metal, which can be used for the degradation of dyes.

Alam U et al. [200] synthesized a Bi-doped $TiO_2NT$/graphene composite catalyst by the hydrothermal method. The synergic effect of Bi and graphene embedded in $TiO_2$ nanotubes promoted the interface charge transfer and improved the visible light efficiency. Wang Weikang et al. [201] synthesized a boron-doped $TiO_2$ (a/r adjustable) photocatalyst by the one-step roasting method. The electron transfer occurring in the two-phase interface is conducive to charge separation, and the charge trap is provided by the b-less electron structure, which improves the degradation capacity of atrazine. Xing Huan et al. [202] prepared a $Ti^{3+}$ self-doped $TiO_2$ single crystal with oxygen vacancy as the electron trap. The results showed that the photodegradation activity of phenol increased significantly. Nair S B et al. [203] prepared self-doped $TiO_2$ nanotubes (TONT) by electrochemical reduction, introduced $Ti^{3+}$ ions, and oxygen vacancy to reduce the band gap, and the degradation of methylene blue reached 97% under visible light irradiation.

Combination is also an easy way to improve the activity of nano-$TiO_2$ in the early stage by combining the appropriate semiconductor to improve the activity of the catalyst.

Liu Chao et al. [204] obtained graphene-like carbon plane grafted g–$C_3N_4$ by calcining, and then conjugated it with $TiO_2$ to construct a ternary heterostructure (carbon plane/g–$C_3N_4$/$TiO_2$), which can be extended to visible light absorption and degrade methylene blue (98.6%), tetracycline (94.0%), and norfloxacin (95.3%), and has good cyclic stability. The carbon plane of the graphene layer makes full contact with the ternary heterojunction, improves the efficiency of charge separation, and inhibits photocatalytic electron-hole pair recombination.

In addition to degrading dyes, $TiO_2$ is also used to degrade other pollutants such as antibiotics, heavy metals, and toxic gases. Pengzhao Ya et al. [205] reported that based on the hard-soft acid-base (HSAB) principle, $TiO_2$ was directly compounded with $NH_2$–UiO–66 to synthesize a $TiO_2$@$NH_2$–UiO–66 nanocomposite used for the photocatalytic degradation of styrene. Due to the porosity of MOF, the quaternary ammonium salt around $TiO_2$ can be well encapsulated, thereby prolonging its residence time at the photocatalytic active site and improving the activity. This method can form a good contact interface between $TiO_2$ and $NH_2$–UiO–66, and can effectively promote the separation of photogenerated electrons and holes. Compared with blank $TiO_2$ and $NH_2$–UiO–66, the degradation efficiency is significantly improved. The $TiO_2$@$NH_2$–UiO–66 photocatalyst has good stability, and the photocatalytic activity can last 600 min without obvious deactivation. Haibo Sheng et al. [206] prepared $TiO_2$@MIL-101 double-shell hollow particles for the photocatalytic degradation of $H_2S$, showing good activity. The conversion rate of hydrogen sulfide can reach up to 90.1%, which is obviously improved when compared with hollow and blank $TiO_2$. The reaction reached equilibrium in 60 min, which is a significant improvement compared to hollow $TiO_2$. Due to its excellent $H_2S$ adsorption capacity, MIL-101 can significantly enhance the photocatalytic activity.

In addition to the above methods, Table 2 lists common ways to improve the degradation activity of photocatalytic dyes, which are similar to those in photocatalytic hydrogen production. Since the dyes have no energy level requirements for other applications, the implementation is relatively simple, and it can be seen that nanotubes and nanometers $TiO_2$ have advantages. From the perspective of activity, the $[Pt_3(CO)_6]_6^{2-}$–$TiO_2$ photocatalyst with noble metal Pt had the highest activity, but due to different degradation dyes and different final degradation rates, it was difficult to compare the difference in activity. From a rough perspective, $TiO_2$ doped with a noble metal such as Ag and Pt have a high activity, but has a high price. Transition metal doping showed higher activity than non-metal doping, showing an advantage in dye degradation. The composites showed high degradation rates, but it took a long time to degrade.

**Table 2.** Summary of common methods to improve $TiO_2$ photocatalytic dye degradation activity.

| Catalysts | Morphology | Reaction Conditions | Catalyst Dosage | Degradation Efficiency for Organic Dyes | Reference |
|---|---|---|---|---|---|
| $Ag^0$–$TiO_2$ nanosol | Nanoparticle | Visible light | 50 mL | 50 mL of $1 \times 10^{-5}$ M RB, 90% in 4 h. | [207] |
| Ag–$TiO_2$ | Nanotube | UVA(360nm) | NA | 3 mL of $2.5 \times 10^{-5}$ M AO7, 80% in 1 h | [208] |
| Au–$TiO_2$ | Nanotube | UVA (360 nm) | NA | 3 mL of $2.5 \times 10^{-5}$ M AO7, 67% in 1 h | [208] |
| Ag–$In_2O_3$–$TiO_2$ | Nanoparticle | UVB (313 nm) | 1.67 g $L^{-1}$ | 90 mL of 25 mg $L^{-1}$ RB, 100% in 45 min | [209] |
| $PtCl_4{}^{2-}$–$TiO_2$ | Nanoparticle | Visible light | 0.5 g $L^{-1}$ | $1 \times 10^{-4}$ M RB, 90% in 2 h | [210] |
| $PtCl_6{}^{2-}$–$TiO_2$ | Nanoparticle | UV–Vis | 0.5 g $L^{-1}$ | $1 \times 10^{-4}$ M RB, 100% in 20 min. | [210] |
| $[Pt_3(CO)_6]_6{}^{2-}$–$TiO_2$ | Nanoparticle | UV–Vis | 0.5 g $L^{-1}$ | $1 \times 10^{-4}$ M RB, 100% in 15 min, | [210] |
| $Fe^{3+}$–$TiO_2$ | Nanoparticle | Visible light | 0.33 g $L^{-1}$ | 15 mL of $1 \times 10^{-7}$ M SRB, 60.5% in 90 min | [211] |
| $Cu^{2+}$–$TIO_2$ | Nanotube | UV | 5 g $L^{-1}$ | 100 mL of 3 mg $L^{-1}$ RB, 97.5% in 50 min. | [212] |
| Zn- $TiO_2$ | Nanoparticle | UV | 1 g $L^{-1}$ | 700 mL of 20 mg $L^{-1}$ MO, 100% in 30 min. | [213] |
| $Cr^{3+}$–$TiO_2$ | Nanotube | UV–Vis | 0.5 g $L^{-1}$ | 100 mL of 20 mg $L^{-1}$ MO, 96.9% in 3 h. | [214] |
| C–$TiO_2$ | Nanotube | Artificial solar light | NA | 97.3% of MB was achieved in 7 h. | [215] |
| N–$TiO_2$ | Nanofilm | Visible light | NA | 47.2% degradation of 30 mL of 20 mg $L^{-1}$ MB | [216] |
| B/N–$TiO_2$ | Nanoparticle | UV–Vis | NA | For RB, 86.5% in 1.5 h | [217] |
| N/S–$TiO_2$ | Nanoparticle | UV | 125 mg $L^{-1}$ | 800 mL of 10 mg $L^{-1}$ MO, 88% in 1.5 h | [218] |
| CdO/ZnO–$TiO_2$ | Nanofilm | Visible light | NA | 500 mL of 100 mg $L^{-1}$ textile blue azo dye, 100% in less than 2 h | [219] |
| $RuO_2$-$SiO_2$–$TiO_2$ | Nanoparticle | UVA (350nm) | NA | 4 mL of 5 mg $L^{-1}$ MO, 100% in 2 h | [220] |
| Ag/$InVO_4$–$TiO_2$ | Nanofilm | Visible light | NA | 30 mL of 10 mg $L^{-1}$ MO, 45% in 15 h. | [221] |

## 5. Photocatalytic Reduction of Carbon Dioxide

Over the past few decades, the massive increase in $CO_2$ and the focus on energy supply are considered as the greatest challenges of this century. Converting $CO_2$ into renewable fuels through artificial photosynthesis has been considered as the best way to avoid both energy and environmental problems. Since Inoue [222] reported that semiconductors can reduce $CO_2$ in water, efforts have been made to build more efficient and environmentally friendly photocatalysts for $CO_2$ reduction. Up to now, most photocatalytic $CO_2$ reduction reactions using metal oxide semiconductor photocatalysts have been conducted under ultraviolet light or high power light irradiation, which is not suitable for practical production [223,224].

Photocatalytic reduction of $CO_2$ is similar to natural photosynthesis, in which plants convert $CO_2$ and $H_2O$ to oxygen and carbohydrates under sunlight. In this process, solar energy is transformed and stored in the form of carbohydrates. It is a combination of water oxidation reaction and carbon dioxide reduction reaction (or $CO_2$ fixation reaction), involving light and dark reaction [225].

Due to the difference in obtained products, the difficulty of photocatalytic carbon dioxide reduction is different, but in order to improve the activity of the $TiO_2$ photocatalyst, most studies still adopt the simple strategy of metal or nonmetal doping or composites.

Andreu et al. [226] applied Mg loaded on the surface of $TiO_2$ (rutile type, anatase type, plate titanium type) nanoparticles for the photocatalytic reduction of $CO_2$, and the products were CO and $CH_4$. A little Mg can increase the concentration of $Ti^{3+}$ (as an electron trap) and modified oxygen (as a hole trap), thereby reducing the charge recombination rate and increasing the activity of the catalyst. Tseng et al. [227] prepared a $TiO_2$-supported Cu catalyst by the sol-gel method, and orderly

modified Cu distribution on the catalyst surface after post-treatment. Under ultraviolet light, the $CH_3OH$ produced by the system was the greatest (1000 $\mu mol \cdot g^{-1}$). Dispersed Cu(I) is considered as a key site for catalytic reduction. When Cu(I) changes to Cu(0) or its aggregates, the catalytic activity of the system decreases.

Xie et al. [228] supported different metal oxides (MgO, SrO, CaO, BaO, $La_2O_3$, $Lu_2O_3$) on the surface of a $TiO_2$/Pt catalyst. The study showed that the content of $CH_4$ for the MgO–$TiO_2$ composite was the highest. After the $CO_2$ molecule is absorbed in MgO, it becomes more concentrated, its structure is more unstable, and it is easier for it to participate in the reaction. Using $TiO_2$ (P25) as the framework, Li et al. [229] synthesized the CuO/$TiO_2$ catalyst (CuO mass fraction was 32%) by the dipping method. Using $Na_2SO_3$ as a sacrificial agent, the yield of reduced products $CH_3OH$ and $C_2H_5OH$ in water was 12.5 and 27.1 $\mu mol/gk$, respectively.

The most important problem limiting the widespread use of $TiO_2$ is the recombination of photogenerated electrons and holes. At present, the heterogeneous structure strategy has been proposed to inhibit recombination [230,231]. However, a single heterojunction had almost no effect on facilitating the separation of electrons and holes. Therefore, the construction of multiple heterostructures is necessary for the collaborative improvement of interface electron–hole separation and migration. Recently, a catalyst involving the design of an m–s junction with a p–n junction has been reported [232]. Compared with a single junction, the interaction of multiple heterojunctions results in the synergistic improvement of photocatalytic activity. As the distribution of the multiple heterogeneous structures on the surface of the photocatalyst is irregular, the effect of photogenerated electrons and holes is still poor. It has been reported that, on account of the difference in the electronic structure of anatase $TiO_2$ {101} and {001} faces, heterostructure can be carried out in different planes to separate the electrons and holes in space and aggregate on the {101} and {001} faces, respectively [223]. Therefore, Aiyun Meng et al. [233] designed one to construct the p–n junction with $MnO_x$ on the $TiO_2$ {001} surface to facilitate hole migration from $TiO_2$ to the p-type semiconductor, and the m–n junction with Pt on the $TiO_2$ {101} surface for electron migration from $TiO_2$ to metal. The synthesized catalyst had a good hole–electron pair separation capability and a high photocatalytic activity. After three hours of light exposure, the maximum yield of $CH_4$ and $CH_3OH$ reached 104 and 91 $\mu mol\ m^{-2}$, respectively, which was higher than that of pure $TiO_2$ and $TiO_2$/Pt, and the catalyst had good stability (Figure 14).

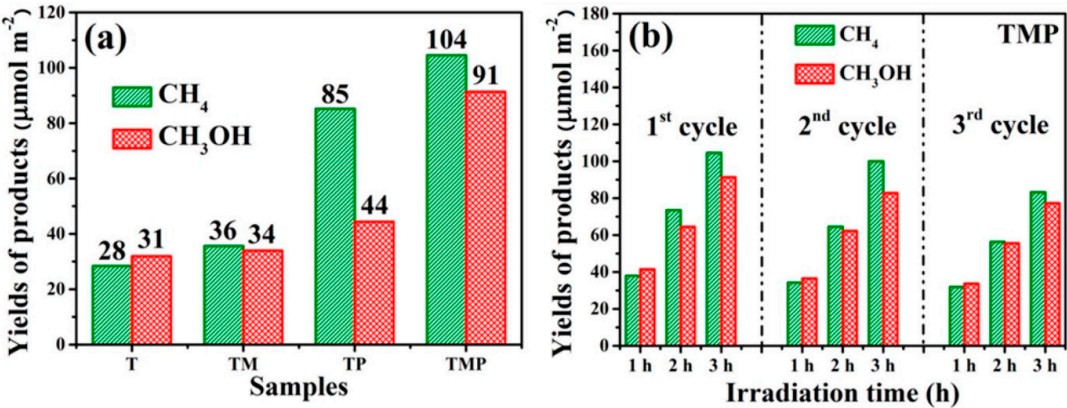

**Figure 14.** (**a**) Photocatalytic degradation of $CO_2$ activity of $TiO_2$ (T), $TiO_2$–Pt (TP), $TiO_2$–$MnO_x$ (TM), and $TiO_2$–$MnO_x$–Pt (TMP) under UV–Vis for 3 h. (**b**) Photocatalytic cycling test of TMP. Reprinted with permission from [233].

Photocatalyst surface modification is an effective method to control the selectivity of photocatalytic degradation of carbon dioxide [234,235]. For example, the hydrophobic catalyst can inhibit the release reaction of hydrogen and improve the product selectivity of $CO_2$ conversion [236]. Sunil et al. [237] reported the successful synthesis of a $Cu_2O$/$TiO_2$ photocatalyst modified with taurine and ethylenediamine to degrade $CO_2$. Compared with the blank catalyst, taurine treatment improved

the selectivity of $CH_4$ on the photocatalyst surface, while ethylenediamine treatment improved the selectivity of CO on the photocatalyst surface. Although the same $Cu_2O/TiO_2$ photocatalyst was used, the product selectivity was significantly changed. Due to the difference in the number of electrons produced in the reaction, for example, different numbers of electrons are needed for $CO_2$ to degrade both CO and $CH_4$, and the ligand-treated samples (*TAU and *EDA) showed an overall enhanced photocatalytic activity. The selectivity of the products of $CH_4$ and CO was quite different from that of the blank photocatalyst. As shown in Figure 15, the $CH_4$ produced by *TAU was 2.4 times that of the blank sample, and the CO was 3.3 times less that of the blank sample; while the CO production rate of *EDA increased by more than 2.0 times, and the production of $CH_4$ was even less than that of the blank sample.

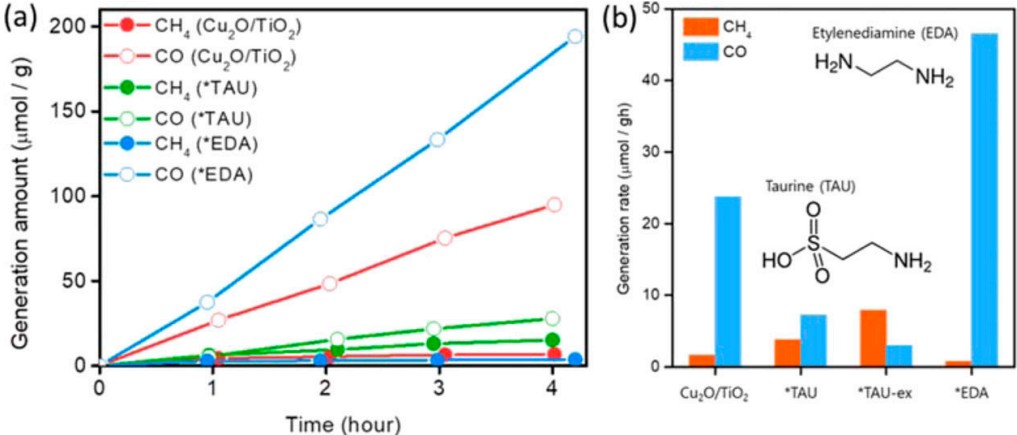

**Figure 15.** (**a**) CO and $CH_4$ are generated in the process of photocatalytic reduction of $CO_2$ for 4 h. (**b**) Photocatalytic degradation of $CO_2$ activity with different ligand treatment of $TiO_2/Cu_2O$. Reprinted with permission from [237].

Photocatalytic degradation of $CO_2$ is a multiple step process including adsorption, $CO_2$ activation, and $CO_2$ bond dissociation [238,239]. However, the reported photocatalytic composites for $CO_2$ reduction showed low $CO_2$ adsorption. Activated carbon fibers (ACFs) have recently been identified as an aussichtsreich large surface area base material that can effectively adsorb $CO_2$ [240,241]. Ajit et al. [242] synthesized a $NiO/TiO_2$ photocatalyst supported on an activated carbon fiber (ACF) (Figure 16). The layered porous structure and large surface area of ACF can better absorb $CO_2$. $NiO/TiO_2$ provides a catalytic surface for the photocatalytic degradation of $CO_2$ under visible light irradiation. The sol-gel prepared $Ni^{2+}$ doped in $TiO_2$ to form $Ti^{3+}$ and oxygen vacancy, and the doping changed the electronic structure of $TiO_2$, thus significantly improving the photoactivity of $TiO_2$ under visible light. The $TiO_2$ and NiO p–n equilibrium connection formed an internal electric field, which effectively improved the separation of photogenerated electrons and holes under visible light, making the photogenerated electrons move to the conductive band of n–$TiO_2$ and the holes to move to the valence band of p–NiO. Electron and hole pairs migrate to the surface of $NiO/TiO_2$, participate in the reduction and oxidation process, and catalyze the degradation of $CO_2$ to generate methanol. In the process of the photocatalytic degradation of $CO_2$ into methanol, the photocatalytic performance was relatively high. The yield of methanol generated under visible light for 2 h was 986.3 $\mu mol\ g^{-1}$. Even after 10 cycles, $NiO/TiO_2/ACF$ was still stable.

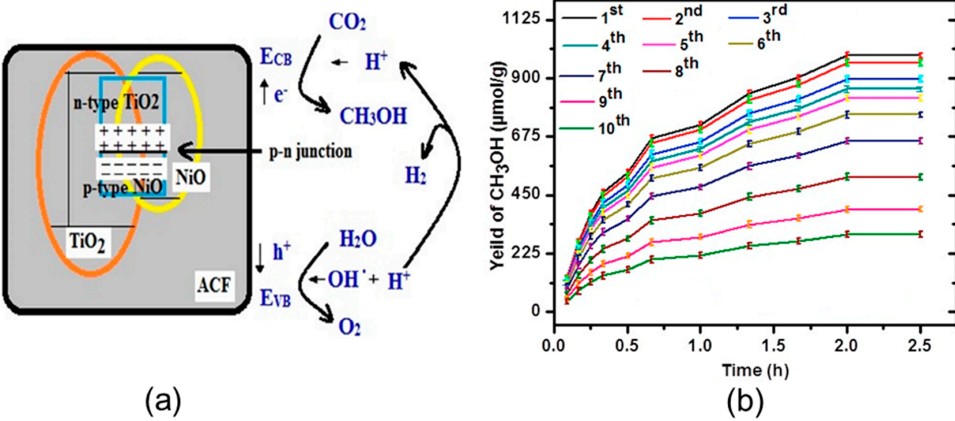

**Figure 16.** (**a**) The mechanism of photocatalytic reduction of $CO_2$ into methanol. (**b**) Photocatalytic cycling test of NiO–$TiO_2$/ACF under visible light irradiation. Reprinted with permission from [242].

According to reports, graphene oxide (GO)/reduced graphene oxide (rGO) exhibits superior electron mobility, high specific surface area, and an adjustable band gap with semiconductor properties. Due to its semiconductor properties, Hsu et al. [243] reported that the reduced carbon dioxide was converted to methanol using GO as the photocatalysts. In this regard, there have been many reports on the functionalization of GO/rGO in the photocatalytic reduction of $CO_2$ into hydrocarbons under visible light [244–246]. Yalavarthi et al. [247] synthesized $TiO_2$ nanotubes coated with GO/rGO for the photocatalytic reduction of $CO_2$ (Figure 17). $TiO_2$ is wrapped in the GO/rGO layer, which also forms an interconnection bridge between adjacent nanotubes. This unique structure can favor photoproduction electron–hole separation and implement effective charge transfer, thus improving the photocatalytic activity. rGO/$TiO_2$ nanotubes showed the highest photocatalytic activity at 2 h of reaction rate of 760 $\mu$mol g$^{-1}$.

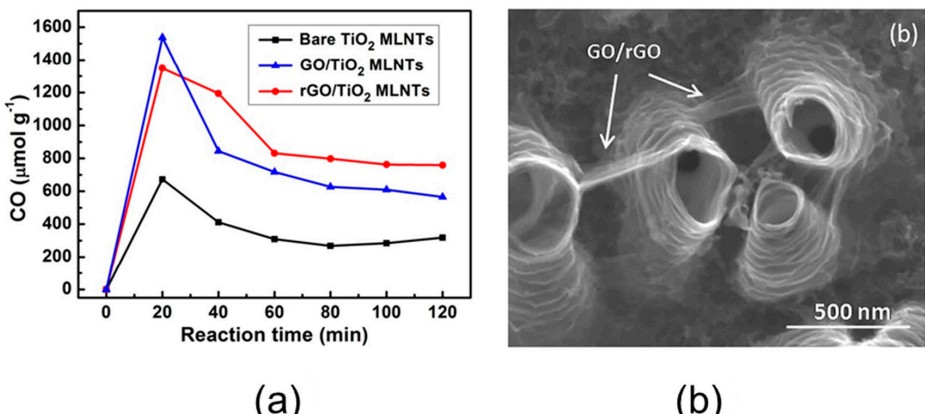

**Figure 17.** (**a**) The amount of carbon monoxide produced by the photocatalytic reduction of carbon dioxide from different composite materials. (**b**) Field Emission Scanning Electron Microscope (FESEM) images of rGO/$TiO_2$ nanotubes. Reprinted with permission from [247].

Table 3 lists the activity and morphology for photocatalytic $CO_2$ reduction. Different products of photocatalytic $CO_2$ reduction require different numbers of electrons, leading to different activity. Most of the morphologies are nanoparticles, and subsequent research should focus on changing the morphologies such as thinner structures with nanocrystals and nanowires, so that electrons can quickly migrate to the surface and participate in the reaction. The process of $CO_2$ degradation to $CH_4$ is the most difficult to achieve and requires the most electrons to participate in the reaction. In–$TiO_2$ exhibited the highest catalytic activity, demonstrating the superiority of metal doping in the catalytic

formation of $CH_4$, and $C–TiO_2$ also had a high catalytic activity to generate HCOOH. At present, $CO_2$ degradation should focus on improving selectivity and generating $CH_4$ with high value-added. However, due to the high energy level of the reaction, such a target is difficult to achieve and the activity will not be particularly high.

**Table 3.** Summary of common methods to improve the photocatalytic reduction of the carbon dioxide activity of $TiO_2$.

| Catalysts | Morphology | Reaction Conditions | Major Product | Product Yield | Reference |
|---|---|---|---|---|---|
| $C–TiO_2$ | Nanoparticle | UV | HCOOH | 438.996 µmol $g^{-1}$ $h^{-1}$ | [248] |
| $N–TiO_2$ | Nanoparticles | UV–Vis | $CH_3OH$ | 10 µmol $g^{-1}$ $h^{-1}$ | [249] |
| $N–TiO_2$ | Nanoparticle | Visible light | $CH_4$ | 0.155 µmol $g^{-1}$ $h^{-1}$ | [250] |
| $V–TiO_2$ | Nanoparticle | Visible light | $CH_3OH$ | 1.15 µmol $g^{-1}$ $h^{-1}$ | [251] |
| $Cr–TiO_2$ | Nanoparticle | Visible light | $CH_3OH$ | 0.25 µmol $g^{-1}$ $h^{-1}$ | [251] |
| $Co–TiO_2$ | Nanoparticle | visible light | $CH_3OH$ | 1.63 µmol $g^{-1}$ $h^{-1}$ | [251] |
| $In–TiO_2$ | Nanoparticle | UV | $CH_4$ | 243.75 µmol $g^{-1}$ $h^{-1}$ | [252] |
| $Ag–TiO_2$ | Nanorod | UV | $CH_4$ | 2.64 µmol $g^{-1}$ $h^{-1}$ | [253] |
| $Pt–TiO_2$ | Nanorod | UV | $CH_4$ | 2.5 µmol $g^{-1}$ $h^{-1}$ | [254] |
| $Au–TiO_2$ | Nanoparticle | UV | $CH_4$ | 1.33 µmol $g^{-1}$ $h^{-1}$ | [255] |
| $CdS/TiO_2$ | Nanoparticle | Visible light | $CH_4$ | 0.188 µmol $g^{-1}$ $h^{-1}$ | [256] |
| $CuPc–TiO_2$ | Nanoparticle | UV–Vis | HCOOH | 26.06 µmol $g^{-1}$ $h^{-1}$ | [257] |

## 6. Photocatalytic Nitrogen Fixation

Ammonia ($NH_3$) is one of the most important products in today's chemical industry. It is the key raw material for the synthesis of various nitrogen-containing compounds [258]. Through the Haber–Bosch process invented in the early 20th century, $NH_3$ is produced by the hydrogenation of $N_2$, which consumes 2% global energy by humans each year and releases large amounts of greenhouse gases ($CO_2$ produced when fossil fuels are burned to provide heat) into the atmosphere [259]. Introducing solar or electrical energy instead of heat into nitrogen fixation reduces energy consumption and greenhouse gas emissions [260]. Among them, due to photocatalytic nitrogen fixation realized under mild conditions, it is considered as one of the most advanced ammonia synthesis methods. Schrauzer et al. [261] first reported $TiO_2$ as a photocatalyst for the reaction. Subsequently, it has been demonstrated that a variety of semiconductors including $Fe_2O_3$ [262], $WO_3$ [239], and BiOBr [263] have photocatalytic nitrogen fixation activities.

Photocatalytic nitrogen fixation is a newly developed branch in recent years, and there are few studies available. Currently, most studies are focused on doping $TiO_2$ to obtain higher performance. Of course, composites are also an appropriate strategy for nitrogen fixation. Weirong Zhao et al. [264] successfully prepared Fe-doped $TiO_2$ nanoparticles with high (1 0 1) face ratio by the two-step hydrothermal method. When ethanol is used as a hole-trapping agent, quantum yield can reach $18.27 \times 10^{-2}$ $m^{-2}$, which is 3.84 times higher than blank $TiO_2$. $Fe^{3+}$ ions uniformly mixed into anatase crystal and replaced $Ti^{4+}$ in $TiO_2$ lattice can increase the density of charge carrier concentration to improve nitrogen fixation activity. Daimei Chen et al. [265] prepared three codoped nanoparticles of C-doped $TiO_2$, N-doped $TiO_2$, and C/N-doped $TiO_2$ by the simple sol-gel method. It is found that doping C and N atoms can inhibit the crystal growth of $TiO_2$. The influence of C doping is more obvious than that of N doping. N atoms can replace oxygen atoms in the lattice of anatase, while most C atoms are deposited on the surface. The results of nitrogen fixation showed that the $C–N–TiO_2$ nanomaterial showed the highest photocatalytic activity. Liu et al. [266] used a MXene derivative to synthesize the $TiO_2@C/g–C_3N_4$ heterojunction, and has rich surface defects, high electron donating ability, appropriate light capture, excellent charge transfer, and the strong ability of nitrogen activation. Excellent optical performance of the catalytic reduction of ammonia nitrogen is thus obtained, and $NH_3$ formation rate was 250.6 µmol $h^{-1}$ $g^{-1}$ under Vis.

The presence of defects in semiconductor photocatalysts has attracted extensive attention as an active site for reactions [267]. Recent studies have shown that a defective catalyst can be activated at a wavelength of 500 nm. It was proven that a thin catalyst could effectively improve the separation of the electron and hole [268]. Yunxuan Zhao et al. [269] synthesized Cu-doped $TiO_2$ nanosheets by using the Jahn–Teller distortion strategy, thus introducing a large number of oxygen vacancies and showing a wide range of solar absorption (Figure 18). The experiment showed that the $TiO_2$ nanosheets doped with Cu had a higher hole–electron separation efficiency and a higher nitrogen fixation activity. $TiO_2$ nanosheets doped with 6% Cu showed an activity of 78.9 $\mu mol\ g^{-1}\ h^{-1}$ under full light irradiation, which was 1.54 and 0.72 $\mu mol\ g^{-1}\ h^{-1}$ under 600 nm and 700 nm monochromatic excitation, respectively.

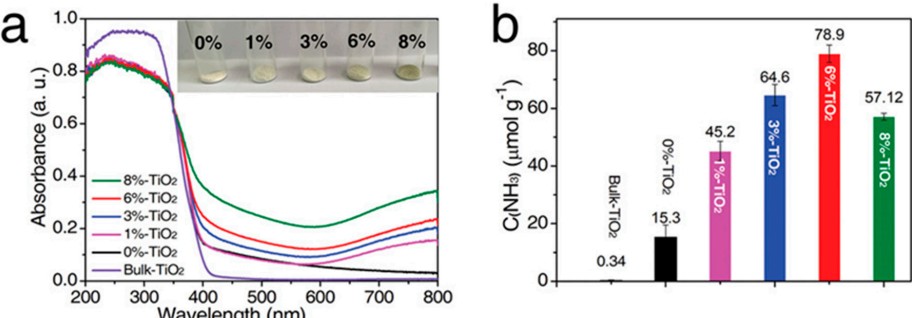

**Figure 18.** (**a**) UV-DRS of X%-TiO2 nanosheets (X = 0, 1, 3, 6, 8) and Bulk-$TiO_2$ (the inset shows a photograph of the X%-$TiO_2$ nanosheets). (**b**) Photocatalytic nitrogen fixation activity of $TiO_2$ with different doping concentrations under UV–Vis after 1 h. Reprinted with permission from [269].

Great efforts have been made to narrow the semiconductor band gap and reduce the photogenic electron–hole recombination rate [270–272]. Meanwhile, morphology adjustment has also been proven to be an effective means to improve photocatalytic activity [273]. For example, photocatalysts for core-shell structured nanoparticles exhibit the ability to modulate light propagation due to their unique morphology. Nanotubes with hollow structure and high specific surface area have higher light utilization due to multiple scattering effects. Shiqun Wu et al. [274] reported a photocatalyst with a high specific surface area and a large number of defects in hollow $TiO_2$ nanotubes. These two features in favor of the capture of light and photo-induced electron and hole separation, so as to promote the photocatalytic nitrogen fixation activity. First of all, the hollow is good for collecting light. Second, rich defects on $TiO_2$ nanotubes (oxygen vacancy and $Ti^{3+}$) with larger surface area (344 $m^2/g$), are the active sites of nitrogen adsorption to improve the photocatalytic activity (Figure 19). Under the light, the titanium dioxide nanotubes show high and stable production of ammonia (106.6 $\mu molg^{-1}\ h^{-1}$).

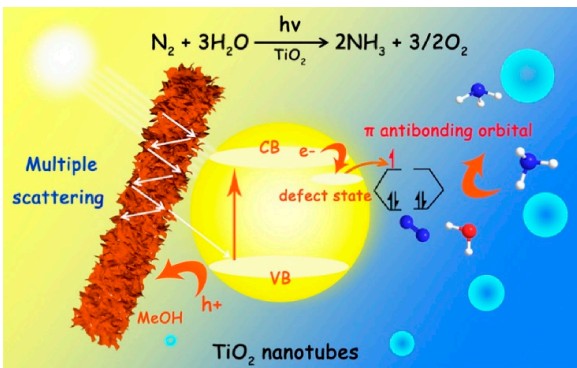

**Figure 19.** Photocatalytic nitrogen fixation mechanism by $TiO_2$ (B) nanotubes. Reprinted with permission from [274].

Recently, Li et al. [275] found that the oxygen vacancy ($O_{vac}$) in BiOBr nanosheets has the ability to absorb and reduce $N_2$ to $NH_3$. Similarly, $O_{vac}$ in $TiO_2$ has the same capacity in the adsorption and reduction of $N_2$, which can be illustrated by the photocatalysis of $N_2$ under ultraviolet light by the $TiO_2$ photocatalyst containing $O_{vac}$ [276]. However, when methods such as $H_2$ reduction are used to introduce $O_{vac}$, they cannot avoid the formation of body defects, which will cause photogenic electron–hole pair recombination [277]. Jiangpeng Wang et al. [278] reported that $TiO_2$ nanotubes with less oxygen vacancy were prepared by treating hydrogenated $TiO_2$ with dicyandiamide (DA) to repair the body defects. Electron spin resonance (ESR) results showed that the sample exhibited a strong paramagnetic peak due to a large number of oxygen vacancies and body defects after hydrogen treatment. However, after the DA treatment of the sample, only a very low ESR peak could be observed, indicating that a large number of body defects had been repaired, leaving only some oxygen vacancies. The $NH_3$ yield of the blank $TiO_2$ nanotubes for photocatalytic $N_2$ fixation was 0.14 mmol·$L^{-1}$·$h^{-1}$, the $N_2$ fixation performance of $TiO_2$–$H_2$–DA was greatly improved, and the generation rate of $NH_3$ reached 1.2 mmol·$L^{-1}$·$h^{-1}$(Figure 20). This value is more than eight times that of original $TiO_2$. It can be seen clearly that oxygen vacancy can improve the activity, but body defects will also be introduced. Appropriate treatment methods will effectively reduce the body defect and reserve part of the oxygen vacancy to improve the activity.

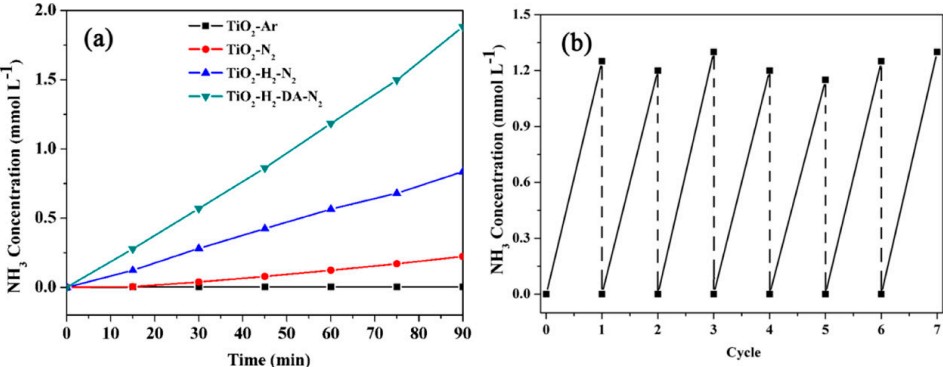

**Figure 20.** (**a**) Photocatalytic nitrogen fixation activities of different photocatalysts, (**b**) photocatalytic nitrogen fixation cycle activities of $TiO_2$–$H_2$–DA. Reprinted with permission from [278].

Due to the fact that photocatalytic nitrogen fixation is a newly developed field and is difficult to realize, the modification of $TiO_2$ is currently in the exploration stage. In addition to the above methods, only a few modification methods of $TiO_2$ have been reported in the literature. Table 4 lists the methods used to improve photocatalytic nitrogen fixation in some literatures. Due to the lack of references, it is difficult to compare the advantages of morphology. At present, the dopant activity of metals is relatively low. To obtain high dopant activity, a composite method is needed, but it is complex, and two suitable band semiconductors are required. The direction of photocatalytic nitrogen fixation is still in the exploratory stage, and the problem to be solved in the present research is to obtain high activity by using relatively easy means.

**Table 4.** Summary of common methods to improve the photocatalytic nitrogen fixation activity of $TiO_2$.

| Catalyst | Morphology | Reaction Conditions | Product Yield | Reference |
|---|---|---|---|---|
| Mg–$TiO_2$ | Nanoparticle | UV–Vis | 10.35 μmol $h^{-1}$ $g^{-1}$ | [279] |
| Cr–$TiO_2$ | Nanoparticle | UV–Vis | 2.12 μmol $h^{-1}$ $g^{-1}$ | [280] |
| V–$TiO_2$ | Nanoparticle | UV–Vis | 6.12 μmol $h^{-1}$ $g^{-1}$ | [281] |
| Ce–$TiO_2$ | Nanoparticle | UV–Vis | 4.25 μmol $h^{-1}$ $g^{-1}$ | [281] |
| Ru/$TiO_2$ | Nanosheet | UV–Vis | 3.32 μmol $h^{-1}$ $g^{-1}$ | [282] |
| Au/$TiO_2$ | Nanotube | UV–Vis | 1.04 μmol $h^{-1}$ $l^{-1}$ | [283] |
| $RuO_2$@$TiO_2$–MXene | Nanoparticle | UV–Vis | 425 μmol $l^{-1}$ $g^{-1}$ | [284] |

## 7. Conclusions

Due to its physical structure and good optical properties, titanium dioxide is considered to be a promising semiconductor photocatalyst, while nano-$TiO_2$ has the advantages of large specific surface area and more exposed active sites, so it has better performance than $TiO_2$. The important environmental applications of the nano-$TiO_2$ photocatalyst were highlighted in this review such as hydrogen production, dye degradation, $CO_2$ degradation, and nitrogen fixation. As reviewed here, a number of studies focused on making nano-$TiO_2$ active in the visible light region by various methods such as doping of metals or nonmetals, manufacturing defects, and compounding of other semiconductors. So far, the successful application of nano-$TiO_2$ photocatalysts in visible light has only been on a laboratory scale. Future research should focus on the use of novel nano-$TiO_2$ photocatalysts (doped nano-$TiO_2$ or composite nano-$TiO_2$) for large-scale application.

**Author Contributions:** S.W., Z.D. and D.-H.W. researched the literature and wrote the manuscript; X.C. and J.X. discussed ideas and edited the manuscript. All authors have read and agreed to the published version of the manuscript.

**Funding:** This research received no external funding.

**Conflicts of Interest:** The authors declare no conflict of interest.

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
