# Peer review of "Modified Nano-TiO2 Based Composites for Environmental Photocatalytic Applications"

_catalysts, doi:10.3390/catal10070759_

Round 1

Reviewer 1 Report

The authors describe the photocatalysis of nano-sized TiO2 for H2 production, ,CO2 reduction, dye degradation, N2 reduction. The manuscript is well organized, however, the importance of nano-sized TiO2 is unclear. Is only high specific surface area important for catalysis, or is there any hidden rule on morphology of N-TiO2 to govern the catalysis?

If only high surface area of TiO2 is the reason for high catalytic activity, it may be OK but not appeal to many readers of this manuscript.

Reviewer 2 Report

In the present review, Dan-Hong Wang and co-workers gave a nice overview of the existing literature on the environmental photocatalytic applications of nanostructured TiO2-based composites. The manuscript is organized in five sections (in addition to a short introduction and final conclusions): the first part is dedicated to a general overview of TiO2 structures and properties, including TiO2 nanostructures, together with a description of the main strategies for improving its photoactivity; the second part described the principles of photocatalytic hydrogen production and gave a literature overview of the application of nanostructured TiO2-based composites in this field; the third part focused the attention on the nanostructured TiO2-based photocatalytic dyes degradation; the fourth part instead showed few general concepts about the photocatalytic reduction of CO2, followed by the possible applications of nano-TiO2 photocatalysts; finally, in the last part the authors reviewed the TiO2 application in the photocatalytic nitrogen fixation. The overall work seems interesting, but there are some issues to be addressed: I believe that the manuscript needs major revision prior to become suitable for publication in Catalysts. In particular, my comments are listed below.

1. In the introduction, lines 39-40, authors wrote that, in addition to the textile industry, large amounts of dyes are also used in other fields: hair dye, leather industry, paper industry, photochemical batteries. In addition to them, organic dyes are also very common in luminescent solar concentrators (LSC) technologies: see DOI: 10.1002/slct.201800126; DOI: 10.1016/j.dyepig.2019.108100; DOI: 10.1016/j.dyepig.2020.108368. Therefore, I invite authors to add this further aspect in the paper, by citing the above mentioned references.

2. In several point of the paper, starting from the title, the abstract and the introduction, authors use the term “Nano-TiO2”. However, I believe that they should give a rigorous definition of “Nano-TiO2” at the first point where it appears in the text (i.e., line 52 on page 2).

3. On lines 55-56, authors wrote “However, there are few systematic and detailed descriptions of the environmental applications of Nano-TiO2.”, but they did not add any reference to these reviews. Instead, I believe that it is fundamental citing most of the previous reviews on this topic, such as DOI: 10.1021/ie303468t; DOI: 10.1016/j.jphotochemrev.2012.10.001; DOI: 10.1016/j.jes.2014.09.023; DOI: 10.1016/j.micromeso.2014.09.040; DOI: 10.1016/j.apcatb.2017.12.005.

4. In the first part of the review, i.e. that dedicated to the TiO2 structures and properties, authors reproduced a set of SEM/TEM images of different TiO2 nanostructures in Figures 2-5. As a suggestion, I believe that it could be better to group these 4 micrographs into a single Figure, by indicating each of them with the letters (a), (b), (c) and (d). Moreover, the authors could provide a more detailed description of these types of TiO2 nanostructures in the text.

5. In my opinion, review articles should not only provide a general overview of a specific topic, but they should also provide a critical point of view. Although the overview provided by Dan-Hong Wang and co-workers is certainly useful and correct, I believe an authoritative critical point of view is here missing. Therefore, at the end of each section dedicated to Nano-TiO2 applications (i.e. photocatalytic hydrogen production, photocatalytic dyes degradation, photocatalytic reduction of CO2, photocatalytic nitrogen fixation), or also where the authors deem it necessary, a comparison of the performances of the various TiO2 photocatalytic systems should be done, in order to identify the best systems for each kind of environmental application. For this purpose, I could suggest authors to add summary tables for each application (one for the photocatalytic hydrogen production, one for the photocatalytic dyes degradation, one for the photocatalytic reduction of CO2, and one for the photocatalytic nitrogen fixation), where the most promising type of nanostructured TiO2, some characteristic parameters of the application (e.g. hydrogen production rate in the first table, dye degradation rate in the second one, etc.) and the relative reference number are specified. In my opinion, Tables are the best way to compare the performances of the various nanostructured TiO2 systems.

Reviewer 3 Report

The manuscript reports a review on the use of modified nano-TiO2 to be active under visible light for applications in efficient environment photocatalysis.

The topic is of interest and it is in line with the topics of the journal. However, I think this review is quite superficial and only reports data from few papers. There are many reviews about this specific topic in literature, with many examples and great summary of the most recent developments and the future prospective. This is a very broad and common subject, this review does not summarize the last discoveries of the literature. I suggest to the Authors to rethink about it, and broaden up the numbers of paper. The use of table of graphs is particularly useful in reviews.

In addition, the manuscript requires to be reviewed by a native English speaker.

In the present form, the manuscript can not be published. Here you have some suggestions.

Abstract: what do you mean with “compositing”?

Introduction:

Please add more recent references about fossil fuel and other emissions.

Please add a reference after this sentence “It is estimated that nearly 17 to 20 percent of water pollution is related to the textile finishing and dyeing industries”.

I do not agree with this statement: However, there are few systematic and detailed descriptions of the environmental applications of  Nano-TiO2”. I suggest to deep the bibliographic research. It is not clear the aim of this review.

2.Titanium dioxide – an introduction

In the different paragraphs, it is not clear which are the advantages of specific crystal phases and morphologies. It looks like the Authors are reporting a list of characteristics, without explaining properly the differences.

This sentence is not clear “For example, anatase TiO2 nanomaterials usually require solution synthesis or low temperature chemical vapor deposition, high temperature deposition and heating reaction often produce rutile[24].”.

This sentence is not clear: For the TiO2 nanomaterial with zero-dimensional structure, it has an isotropic structure, and all crystal planes will be exposed (including those with higher energy), which is conducive to photocatalytic reactions”.

2.3.1 Metal doping/ 2.3.2 Non-metal doping

In this paragraph the Authors are not reporting which are the main metal dopants used and which ones are better in specific applications. The same is for non-metal doping.

Round 2

Reviewer 2 Report

In this revised version of the manuscript, Dan-Hong Wang and co-workers addressed very satisfactorily all the issues listed in my previous report, thus allowing to improve the quality of the work. I believe that the review now meet the standards for publication in the MDPI Catalysts journal, thus suggesting its acceptance in the present form.

Author Response

Dear Reviewer,

Thank you for your letter and for the reviewers’ comments concerning our manuscript entitled “Modified Nano-TiO2 Based Composites for Environmental Photocatalytic Applications” (review, No. catalysts-822540).

Thank you for your affirmation of our work.

Reviewer 3 Report

I appreciate the Authors' effort to address my comments. 

However, I think the manuscript is still missing a critical view of the topic. It appears to me a list, with no critical comparison of the different technologies. The tables with the summary of the data published in each single study are not helping in this case. The Authors should have clustered similar work together, creating connections and comparison.

For these reason, I think the manuscript is not ready to be published.

Round 3

Reviewer 3 Report

I think the manuscript is better organized now. I only suggest a revision by a native English speaker. 

Author Response

Dear Editors and Reviewers:

Thank you for your letter and for the reviewers’ comments concerning our manuscript entitled “Modified Nano-TiO2 Based Composites for Environmental Photocatalytic Applications” (review, No. catalysts-822540). We have studied the comments carefully and have made corrections which we hope meet with approval.

The manuscript have been a revision by a native English speaker.
